# Thompson Sampling via Fine-Tuning of LLMs

**Nicolas Menet**[1,2]**, Aleksandar Terzić**[1,2]**, Michael Hersche**[1]**, Andreas Krause**[2]**, Abbas Rahimi**[1]

[1]IBM Research – Zurich, [2]Department of Computer Science, ETH Zürich

## Abstract

Bayesian optimization in large unstructured discrete spaces is often hindered by the computational cost of maximizing acquisition functions due to the absence of gradients. We propose a scalable alternative based on Thompson sampling that eliminates the need for acquisition function maximization by directly parameterizing the probability that a candidate yields the maximum reward. Our approach, *Thompson Sampling via Fine-Tuning* (ToSFiT), leverages the prior knowledge embedded in prompt-conditioned large language models, and incrementally adapts them toward the posterior. Theoretically, we derive a novel regret bound for a variational formulation of Thompson Sampling that matches the strong guarantees of its standard counterpart. Our analysis reveals the critical role of careful adaptation to the posterior probability of maximality—a principle that underpins our ToSFiT algorithm. Empirically, we validate our method on three diverse tasks: FAQ response refinement, thermally stable protein search, and quantum circuit design. Within a collection of methods covering in-context Bayesian optimization, reinforcement learning, and evolutionary search, ToSFiT exhibits both state-of-the-art sample efficiency and computational efficiency. [1]

## 1 Introduction

Humans rely on beliefs, shaped by internal world models, to make decisions. Since our knowledge of the world is often incomplete, these beliefs must account for uncertainty through probabilistic reasoning. To stay effective, beliefs must be updated as new information arises. Bayesian inference provides a principled method for this updating process, using Bayes' theorem to adjust beliefs by combining prior knowledge with new evidence. This foundational idea extends naturally to computational settings, where algorithms must make decisions under uncertainty.

Bayesian optimization (Kushner, 1964; Garnett, 2023) is one such algorithmic framework that leverages Bayesian inference for large-scale experimental design and automated discovery, particularly in settings where experimental evaluations are costly or time-consuming. As a strategy for optimizing expensive black-box reward functions, it maintains a posterior distribution over the unknown rewards—typically modeled as a Gaussian process over a domain $X$—and uses this model to guide the search for promising configurations. New candidates are selected by maximizing an acquisition function that balances two objectives: exploring uncertain regions to gather new information, and exploiting areas that are already known to perform well (Kushner, 1964; Auer, 2002).

Among existing acquisition strategies, Thompson sampling (Thompson, 1933; Russo et al., 2018) stands out due to its state-of-the-art convergence guarantees (Russo & Van Roy, 2014; 2016; Chowdhury & Gopalan, 2017) and its strong empirical performance (Chapelle & Li, 2011), despite being one of the earliest strategies for Bayesian optimization. Thompson sampling draws a function from the reward posterior, which effectively serves as an acquisition function, and selects the point that maximizes it. The resulting evaluation points are distributed according to the probability of maximality (PoM) of rewards (Menet et al., 2025).

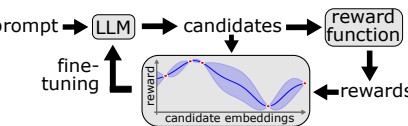

Figure 1: ToSFiT treats candidate generation as Thompson sampling and fine-tunes the LLM to the posterior PoM.

---

[1]Code at https://github.com/IBM/thompson-sampling-via-fine-tuning-of-llms.

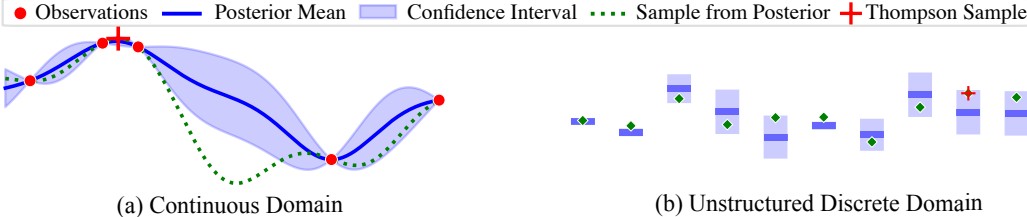

Figure 2: Whereas in continuous domains gradient ascent can be used to maximize the acquisition function, in unstructured discrete domains maximization would require iterating over all points. To scale to combinatorially large discrete domains, we propose to circumvent intractable acquisition function maximization by directly sampling from the parameterized probability of maximality.

While acquisition function maximization has been made tractable in high-dimensional Euclidean spaces—through projection to lower dimensions (Wang et al., 2016; Kirschner et al., 2019) or local optimization (Eriksson et al., 2019)—it remains a fundamental challenge in large unstructured discrete domains, where the absence of gradients precludes efficient search. Yet, such spaces are of high scientific and economic relevance. Notable examples include the space of amino acid sequences (Jumper et al., 2021; Liu et al., 2023) and the space of valid code for quantum circuits (Javadi-Abhari et al., 2024; Vishwakarma et al., 2024). Note that with 20 amino acids and a maximum sequence length of 100, the search space already exceeds the number of atoms in the observable universe.

In this work, we scale Bayesian optimization to large unstructured discrete spaces by instantiating Thompson Sampling as fine-tuning of large language models (TOSFIT, see Figure 1). Our approach builds on Variational Bayesian Optimistic Sampling (VBOS), established by O'Donoghue & Lattimore (2021), but introduces a key distinction: instead of optimizing toward the posterior probability of maximality starting from a uniform policy, we initialize it with a prompt-conditioned pre-trained LLM, and gradually adapt it toward the posterior. By treating generated proposals as Thompson samples, TOSFIT avoids intractable acquisition function maximization. Our contributions are:

1. We improve the cumulative regret bound of exact VBOS from $\tilde{\mathcal{O}}(\sqrt{T|X|})$ to $\tilde{\mathcal{O}}(\sqrt{T\gamma^T})$, accounting for reward correlation across the search space $X$ through the maximal information gain $\gamma^T$, and generalize the bound to cover approximate (e.g., gradient-based) VBOS.

2. Our novel regret bound motivates policy initialization according to pre-training & context as well as deliberate fine-tuning to the posterior PoM, resulting in the TOSFIT algorithm.

3. We validate our method on FAQ response refinement, protein optimization, and quantum circuit design. Within a collection of methods covering Bayesian optimization, reinforcement learning, as well as evolutionary search, ToSFiT exhibits state-of-the-art performance.

## 2 PRELIMINARIES

Consider an unknown reward vector $R$ over a discrete domain $X$. In Bayesian optimization, we want to find the $x \in X$ that maximizes $R_x$. Typically, we model our prior belief as Gaussian, i.e., $R \sim \mathcal{N}(\mu, K)$ with mean $\mu$ and covariance $K$. Thompson sampling suggests conducting Bayesian optimization by repeatedly conditioning the Gaussian reward vector $R$ on a noisy observation $Y_x := R_x + \mathcal{N}(0, \sigma_n^2)$ at a point $x \in X$ sampled from the posterior probability of maximality

$$\text{PoM}(x|\text{data}) := \mathbb{P}[R_x = R^* \mid \text{data}] \text{ with } R^* := \max_z R_z.$$

In practice, sampling from the probability of maximality is typically implemented by drawing a realization from the reward posterior and then selecting the point that maximizes this sample—effectively treating the realization as an acquisition function (Russo et al. 2018, see Figure 2). However, in large unstructured discrete domains, such maximization becomes intractable. To address this, we propose to directly parameterize the probability of maximality using a generative LLM. This approach, illustrated in Figure 1, avoids explicit maximization and, following the framework of O'Donoghue & Lattimore (2021), ensures consistency with the underlying reward model and thus vanishing regret (see Theorem 1).

## 2.1 Batched Bayesian Optimization

Traditionally, Bayesian optimization is performed sequentially: one candidate is selected, evaluated, and used to update the Gaussian reward model before the next candidate is chosen. However, in many real-world applications—such as drug discovery (Houghten, 2000) or automated program synthesis (Romera-Paredes et al., 2024)—simultaneous candidate evaluation can significantly reduce wall-clock time, motivating the development of batched Bayesian optimization (Ginsbourger et al., 2010; Azimi et al., 2012; Desautels et al., 2014; Wang et al., 2018). Classical Bayesian optimization methods typically propose a single, deterministic candidate, and therefore require additional mechanisms to promote diversity across the batch and avoid redundant evaluations. In contrast, Thompson sampling naturally generates diverse candidates by independently sampling from the posterior (Hernández-Lobato et al., 2017), achieving equal asymptotic regret as in the sequential setting (Kandasamy et al., 2018; Nava et al., 2022). This property makes Thompson sampling particularly well-suited for batched optimization, a key feature of ToSFiT.

## 2.2 Regret Bounds for Bayesian Optimization

Bayesian optimization algorithms such as Thompson sampling and Upper Confidence Bound (UCB, Srinivas et al. 2010) are typically framed as balancing exploration with exploitation (Kushner, 1964; Auer, 2002) such that, assuming the black-box reward function $R : X \rightarrow \mathbb{R}$ is drawn from a Gaussian process, in expectation cumulative regret grows sublinearly:

$$\mathbb{E}[\sum_{t=1}^{T} R^* - R_{x^t}] \in o(T).$$

Here, $x^t$ denotes the action played by the Bayesian optimization algorithm conditioned on a history of observations $\mathcal{H}^t$. Note that sublinear cumulative regret is equivalent to vanishing average regret, i.e., asymptotically optimal performance.

Beyond the Bayesian setting, regret guarantees can also be established for the model-agnostic setting that assumes the reward $r$ to lie in a reproducing kernel Hilbert space $\mathbb{H}$ induced by a kernel $K : X \times X \rightarrow \mathbb{R}_+$.[2] In this case, a Gaussian reward model is still used for inference, but its amplitude (i.e., prior uncertainty) must be chosen large enough to accommodate the complexity of $r$, as measured by the norm $\|r\|_K$. Notable examples of such model-agnostic algorithms include GP-UCB (Srinivas et al., 2010) and GP-TS (Chowdhury & Gopalan, 2017), a variant of Thompson sampling.

In practice, however, hyperparameters such as the amplitude are often fit to observations via marginal likelihood maximization (MLM), which can underestimate the true complexity of the reward function (Berkenkamp et al., 2019). To avoid overconfident posterior estimates and premature convergence, it is common to adopt a multiplicative *exploration bonus* on the MLM-fitted prior amplitude. This ensures sufficient uncertainty for effective exploration—a strategy we incorporate into ToSFiT to account for the fixed-reward setting used in our experiments.

## 2.3 Variational Bayesian Optimistic Sampling

As shown in O'Donoghue & Lattimore (2021) and Tarbouriech et al. (2024), Gaussian PoM is effectively approximated by the VBOS policy $\tilde{\pi}$, the distribution that maximizes the functional

$$\mathcal{V}(\pi) := \mathbb{E}_{x \sim \pi}[\mu_x + \underbrace{\sqrt{-2 \ln(\pi_x)}}_{\text{adaptive UCB exploration bonus}} \cdot \sigma_x], \text{ where } \pi \text{ is a distribution over } X \text{ and } \sigma_x := \sqrt{K_{x,x}}. \quad (1)$$

Here, the reward surrogate $\hat{r}_x := \mu_x + \sqrt{-2 \ln(\pi_x)} \cdot \sigma_x$ represents a high-probability upper bound on $R_x$ (O'Donoghue & Lattimore, 2021). VBOS reveals an intimate connection (Tarbouriech et al., 2024; Menet et al., 2025) between PoM and upper confidence bounds augmented with entropy regularization (Ziebart, 2010; Haarnoja et al., 2018). Moreover, $\max_\pi \mathcal{V}(\pi) \geq \mathcal{V}(\text{PoM}) \geq \mathbb{E}[R^*]$ (see Corollary 2 in Appendix E), which is the key property that allows bounding the expected cumulative

---

[2]The expressivity of a reproducing kernel Hilbert space can vary widely with the choice of its kernel. For instance, the linear kernel induces the reproducing kernel Hilbert space of (all) linear functions, whereas the radial basis function kernel corresponds to a smooth subset of the square-integrable functions $L_2$ that can uniformly approximate any continuous function on a compact set.

regret by $\tilde{\mathcal{O}}(\sqrt{T})$ for any optimistic policy $\pi$ with $\mathcal{V}(\pi) \geq \mathbb{E}[R^*]$. In addition to its variational form, Menet et al. (2025) pointed out a near-closed form with almost-linear runtime in $|X|$, given by

$$\tilde{\pi}_x = v(\tfrac{\mu_x - \kappa^*}{\sigma_x}) \text{ for } v(c) := \exp(-(\sqrt{c^2 + 4} - c)^2/8) \text{ with } \kappa^* \text{ such that } \sum_x \tilde{\pi}_x = 1. \quad (2)$$

In combinatorially large domains $X$, even the near-closed form of Equation (2) becomes too expensive, despite its almost-linear runtime in $|X|$. Instead, an approximate policy $\pi \approx \tilde{\pi} := \arg\max_{p \in \Delta^{|X|-1}} \mathcal{V}(p)$ based on gradient ascent of Equation (1) must suffice. Our theoretical results in Section 4 reveal the challenges of Bayesian optimization with gradient-based policy updates. As a remedy to accelerate convergence, we cast Thompson sampling as fine-tuning of LLMs.

## 3 THOMPSON SAMPLING VIA FINE-TUNING

We introduce *Thompson Sampling via Fine-Tuning* (ToSFiT), a scalable variant of Thompson sampling that leverages strong priors from generative pre-training and task-dependent in-context conditioning. ToSFiT does not maximize an acquisition function, but rather parameterizes the PoM with a pre-trained prompt-conditioned large language model (see Figure 1). By considering model generations as samples from the PoM, we avoid expensive acquisition function maximization. To stay consistent with the posterior PoM and achieve sublinear cumulative regret, we initialize the policy according to the pre-training and then cautiously adapt the model parameters using the VBOS objective. This algorithmic design is guided by the theoretical analysis in Section 4.

---

**Algorithm 1** ToSFiT with Gaussian Process Reward Model

---

**Require:** pre-trained policy $\pi^\theta$, GP feature map $\phi$
  Sample $x_1, \ldots, x_m \sim \pi^\theta$ and observe $y_1, \ldots, y_m$
  Conduct GP marginal likelihood maximization
  **while** budget not exhausted **do**
    $\mu^\phi, K^\phi \leftarrow$ closed-form Gaussian posterior
    **for** $j = 1, \ldots, c$ **do**
      Generate $x_1, \ldots, x_B \sim \pi^\theta$
      Estimate $\frac{d}{d\theta} \mathcal{V}_{\mu^\phi, K^\phi}(\pi^\theta)$ using $x_1, \ldots, x_B$
      Fine-tune $\pi^\theta$ toward VBOS with learning rate $\eta$
    **end for**
    Observe $y_1, \ldots, y_b$ associated with $x_1, \ldots, x_b$
    Conduct GP marginal likelihood maximization
  **end while**
  **return** $(x_{\arg\max_i y_i}, \max_i y_i)$

**Hyperparameters** $m \in \mathbb{N}$ denotes a burn-in period to find prior parameters of the GP and allow the closed-form solution $\tilde{\pi}^t$ of the VBOS objective to approach the generative policy $\pi^\theta$ from pre-training (more on that in Section 4). $c \in \mathbb{N}$ is the number of steps of gradient ascent per step of Bayesian optimization and trades off sample efficiency with computational efficiency. $B \in \mathbb{N}$ is the generation batch size: a large batch size improves GPU utilization and leads to more stable gradients. $b \in \{1, \ldots, B\}$ denotes the batch size of Bayesian optimization. $\eta$ is the global learning rate.

---

### 3.1 SCALING GAUSSIAN PROCESSES

The Moore–Aronszajn theorem states that any kernel $K$ can be expressed as an inner product in a reproducing kernel Hilbert space $\mathbb{H}$, i.e., $K(x, z) = \langle \phi(x), \phi(z) \rangle_\mathbb{H}$ for a feature map $\phi : X \to \mathbb{H}$. Thus we assume, without loss of generality, a linear kernel in $\mathbb{H}$, enabling scalable Gaussian process inference. Flexibility is retained through the choice of feature map $\phi$, which can be obtained from fixed embedding models (Ranković & Schwaller, 2023; Wei et al., 2022), learned adaptively during optimization (Ranković & Schwaller, 2025), or designed for the task at hand (Greenhill et al., 2020).

Furthermore, as detailed in Appendix D, we leverage a formulation of linear Gaussian processes that enables conditioning on observations, computing the reward posterior, and performing marginal likelihood maximization—all in closed form. As a result, both the computational and memory complexity scale in $\Theta(\dim(\mathbb{H})^2)$, independently of the number of observations, and has negligible overhead (see Table 2 in Section A.1 of the Appendix).

### 3.2 GRADIENTS OF VARIATIONAL BAYESIAN OPTIMISTIC SAMPLING

We derive explicit gradients of the VBOS objective, offering new insights into its optimization landscape. All proofs are in Appendix E. Note that the score trick (Williams, 1992) cannot be applied blindly, since the reward surrogate $\hat{r}_x := \mu_x + \sqrt{-2\ln(\pi_x^\theta)} \cdot \sigma_x$ depends on the policy $\pi^\theta$.

**Proposition 1.** *Consider the* VBOS *objective* $\mathcal{V}(\pi^\theta) := \mathbb{E}_{x \sim \pi^\theta}[\mu_x + \sqrt{2\ln(1/\pi_x^\theta)} \cdot \sigma_x]$. *The* VBOS *objective $\mathcal{V}$ is concave (strictly if $\sigma_x > 0 \, \forall x$) and its gradients are*

$$\frac{d}{d\theta}\mathcal{V}(\pi^\theta) = \mathbb{E}_{x \sim \pi^\theta}\Big[\big(\mu_x \underbrace{-\xi - v^{-1}(\pi_x^\theta) \cdot \sigma_x}_{-\mu_x^\theta \text{ for } \xi = \kappa}\big) \cdot \frac{d}{d\theta}\ln\pi_x^\theta\Big].$$

$\xi \in \mathbb{R}$ *is an arbitrary baseline and* $-v^{-1}(u) = \sqrt{-2\ln(u)} - 1/\sqrt{-2\ln(u)} \sim \sqrt{-2\ln(u)}$ *as* $u \to 0$.

**Interpretation as Energy-Based Model**  For a given reward uncertainty $\sigma_x$ and (parametrized) probability of maximality $\pi_x^\theta$ there is, according to Equation (2) and up to $\kappa \in \mathbb{R}$, exactly one consistent expected reward given by $\mu_x^\theta := \kappa + v^{-1}(\pi_x^\theta) \cdot \sigma_x$. Gradient ascent on $\mathcal{V}$ thus pushes up the probability of sample generation if and only if $\mu_x^\theta$ underestimates the true $\mu_x$. In this light, VBOS can be considered an energy-based model (LeCun et al., 2006).

### 3.3 Stabilizing Gradients of VBOS

To ensure vanishing regret according to Theorem 1 of Section 4, we must fine-tune the prompt-conditioned pre-trained policy $\pi^\theta$ toward the posterior PoM using gradient ascent on the VBOS objective. Define the pseudo reward $\hat{r}_x^\theta := \mu_x - v^{-1}(\pi_x^\theta) \cdot \sigma_x$ that occurs in Proposition 1. Then

$$\frac{d}{d\theta}\mathcal{V}(\pi^\theta) \approx \frac{1}{B}\sum_i (\hat{r}_{x_i}^\theta - \xi_i) \cdot \frac{d}{d\theta}\ln\pi_{x_i}^\theta \text{ with } x_i \sim \pi^\theta.$$

Here, $\hat{r}_{x_i}^\theta - \xi_i$ is referred to as the advantage function. In practice, this estimator can suffer from high variance, depending on the choice of baselines $\xi_i \in \mathbb{R}$. To address this, we adopt the Reinforce Leave-One-Out (RLOO) baseline (Kool et al., 2019), where each $\xi_i$ is set to the average of the other surrogates in the batch: $\xi_i = \frac{1}{B-1}\sum_{j \neq i} \hat{r}_{x_j}^\theta$. This technique has been demonstrated to outperform more complex alternatives like Proximal Policy Optimization (Schulman et al., 2017) in fine-tuning large language models (Ahmadian et al., 2024). To further stabilize learning, we normalize the advantage function by its empirical standard deviation, derived from the empirical second moments: $\widehat{\text{advantage std}} = \sqrt{\frac{1}{B}\sum_h (\hat{r}_{x_h}^\theta - \xi_h)^2}$, which amounts to a variance-adaptive learning rate. As shown in Proposition 3 of Appendix C, standardized RLOO is mathematically equivalent to the advantage function used in Group Relative Policy Optimization (Shao et al., 2024).

## 4 Theory

Given that one cannot exactly maximize VBOS in practice without incurring prohibitive computational cost, how close is any policy to the maximizer? Geometrically, the divergence of a policy $\pi$ from the exact VBOS policy $\tilde{\pi}$ is measured by the suboptimality gap in the VBOS objective:

**Proposition 2.** *Let $\sigma \in \mathbb{R}_+^{|X|}$. For the convex $f(p) := -\sum_x p_x \sigma_x \sqrt{-2\ln p_x}$, define the Bregman divergence $D_\sigma(p,q) = f(p) - f(q) - \langle \nabla f(q), p - q \rangle$. Then the Bregman divergence of any $\pi \in \Delta^{|X|-1}$ from the maximizer $\tilde{\pi} := \arg\max_{p \in \Delta^{|X|-1}} \mathcal{V}(p)$ is given by $D_\sigma(\pi, \tilde{\pi}) = \mathcal{V}(\tilde{\pi}) - \mathcal{V}(\pi)$.*

Even the exact VBOS policy $\tilde{\pi}$ only conducts approximate Thompson sampling. Nevertheless, O'Donoghue & Lattimore (2021) prove an upper bound on the cumulative regret incurred when sampling a bandit according to exact VBOS, i.e., for $x^t \sim \tilde{\pi}^t$. The bound reads $\mathbb{E}[\sum_t R^* - R_{x^t}] \leq \sqrt{2|X|T\ln|X|(1+\ln T)}$. Note that the structure of the kernel is not taken into account, i.e., the worst-case bandit with independent arms is assumed. In Theorem 1, we demonstrate a significantly tighter regret bound for VBOS, matching the strong regret bounds of Thompson sampling (Russo & Van Roy, 2014) and GP-UCB (Srinivas et al., 2010), regardless of the kernel.

**Theorem 1.** *Let $R \sim \mathcal{N}(\mu, K)$ with $K_{x,x} \leq 1 \, \forall x \in X$ and additive observation noise $\mathcal{N}(0, \sigma_n^2)$.[3] If $R$ is observed at $x^t \sim \pi^t$ for a policy $\pi^t$ depending on history $\mathcal{H}^t$, then*

$$\mathbb{E}[\textstyle\sum_{t=1}^T R^* - R_{x^t}] \leq \sqrt{C_{\sigma_n}HT\gamma^T} + \mathbb{E}\textstyle\sum_{t=1}^T D_{\sigma^t}(\pi^t, \tilde{\pi}^t).$$

$C_{\sigma_n} := 4/\ln(1 + \sigma_n^{-2})$ *is a constant, $H := \frac{1}{T}\sum_t H[\pi^t|\mathcal{H}^t]$ is the expected average entropy of the policy and hence upper bounded by $\ln|X|$, $\gamma^T := \max_{L^T} I(Y_{L^T}; R)$ is the maximum information gain for $T$ observation locations $L^T$, and $\tilde{\pi}^t$ is the unconstrained maximizer of* VBOS *given $\mathcal{H}^t$.*

---

[3] The theorem also holds for heteroscedastic additive Gaussian noise by replacing $\sigma_n$ with $\max_{x \in X} \sigma_n(x)$.

**For exact VBOS**, we have improved the cumulative regret bound from $\tilde{\mathcal{O}}(\sqrt{T|X|})$ to $\tilde{\mathcal{O}}(\sqrt{T\gamma^T})$. Whereas the former is vacuous in combinatorially large discrete domains $X$, the latter is not. E.g., with a linear kernel in $d$ dimensions, we have $\gamma^T \in \mathcal{O}(d \log T)$, yielding a regret bound that scales gracefully with the problem size (Srinivas et al., 2010).

**For inexact VBOS**, we provide the first regret bound. It depends on a Bregman divergence between the exact solver $\tilde{\pi}^t$ of VBOS, and the sampling policy $\pi^t$. As Proposition 2 establishes, this Bregman divergence directly captures to what extent $\pi^t$ maximizes the variational objective $\mathcal{V}$.

## 4.1 INSIGHTS: POLICY INITIALIZATION VIA PRE-TRAINING & CONTEXT

The approximation error $D_{\sigma^t}(\pi^t, \tilde{\pi}^t)$ risks dominating the regret bound, since gradient-based optimization of the VBOS objective $\mathcal{V}$ yields a policy $\pi^t$ that lags behind the exact VBOS policy $\tilde{\pi}^t$. To mitigate this source of cumulative regret, $\pi^t$ must be initialized and maintained within a high-probability neighborhood of $\tilde{\pi}^t$, especially for large $t$, when $\tilde{\pi}^t$ tends to concentrate near the pre-trained policy. To support this, TOSFIT initializes $\pi^t$ via pre-training and prompt-conditioning, and adapts it cautiously

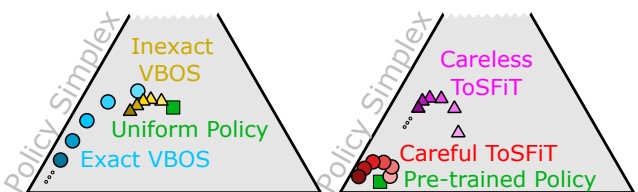

Figure 3: Left: Inexact gradient-based VBOS struggles to find the best arm in very high-dimensional probability simplices. Right: TOSFIT starts in the right region of the simplex. Still, careful adaptation towards VBOS is paramount to retain the prior knowledge necessary to find the best arm.

toward the posterior PoM $\tilde{\pi}^t$ using small learning rates. These insights are illustrated in Figure 3 for a three-armed bandit and experimentally verified in Section 5.6. Note that Theorem 1 holds even if the pre-training prior $\pi^0$ is biased toward suboptimal regions. In that case, more compute effort in the form of gradient steps are required to ensure a vanishing $D_{\sigma^t}(\pi^t, \tilde{\pi}^t)$.

## 5 EXPERIMENTS

We empirically evaluate TOSFIT across three diverse domains for language models between $0.6$B and $8$B parameters. We report the mean and standard error of the *best-seen reward* using $25$ random seeds, defined as the maximum reward observed up to step $t$, i.e., $\max_{t \leq n} r_{x^t}$. This metric captures the best solution found so far and is often used in black-box optimization benchmarks. Details are in Appendix B. Within a collection of methods covering Bayesian optimization, reinforcement learning, and evolutionary search, TOSFIT exhibits state-of-the-art sample efficiency and computational efficiency. TOSFIT can naturally be applied in batched settings, and sample efficiency can be further improved by adjusting the computational effort per round of Bayesian optimization.

## 5.1 SETTINGS

**FAQ Refinement** is a natural language task that tests the algorithm's ability to optimize text based on semantic alignment. We ask a Qwen3-1.7B/Qwen3-8B model (Yang et al., 2025) to write a frequently-asked-questions (FAQ) response. The reward is modeled as the alignment to an unknown ground-truth, judged by the Qwen3-Embedding-0.6B model (Zhang et al., 2025). As a deep kernel, we adopt a linear kernel over the first 256 entries of the embeddings of Qwen3-Embedding-0.6B. The search space consists of all token sequences, which is exponentially large in the response length.

**Protein Search** explores the challenge of designing thermally stable proteins—a task with significant implications for drug development and industrial biotechnology. The goal is to identify amino acid sequences that exhibit high thermal stability, a property that enhances protein robustness and shelf life. Note that with 20 standard amino acids in the human body and sequence lengths of 100 and above, the search space exceeds the number of atoms in the observable universe. We sample amino acid sequences from ProtGPT2 (Ferruz et al., 2022) (0.738B parameters) and score them according to their negative thermal instability index (Guruprasad et al., 1990). Two baselines, FIBO and EVOLUTIONARY SEARCH (LLM), require instruction-tuned models. There, we use a Qwen3-

0.6B/Qwen3-8B model. To predict the thermal stability and assess epistemic uncertainty, we adopt a GP with linear kernel over the mean token embeddings from ProtGPT2 projected to the unit sphere.

**Quantum Circuit Design** is the task of designing quantum circuits that prepare low-energy quantum states in unknown environments. The challenge lies in navigating a vast, discrete space of valid quantum programs, where entanglement and gate structure critically influence performance. To generate Qiskit circuits (Javadi-Abhari et al., 2024), we use a Qwen2.5-Coder-1.5B/Qwen2.5-Coder-7B model (Hui et al., 2024) prompted to fill-in-the-middle (Bavarian et al., 2022) after initializing six disentangled qubits in the zero state. The baselines FIBO and EVOLUTIONARY SEARCH (LLM) require instruction-tuned models, for which we use Qwen2.5-Coder-1.5B-Instruct/Qwen2.5-Coder-7B-Instruct. As reward, we consider the negative energy of the prepared state under an unknown Hamiltonian with strong interaction terms, requiring entanglement for optimal performance. The feature map for the linear Gaussian process over rewards consists of a code validity bit as well as all two-qubit Pauli observables, which can be efficiently simulated using quantum computers.

## 5.2 BASELINES

We compare against seven baselines. For fair comparison, the reward model (if present) is always a Gaussian process over the same (deep) embeddings. As in Li et al. (2022), UNGUIDED GENERATION samples directly from the pre-trained LLM without feedback, serving as a non-adaptive baseline. As in Kristiadi et al. (2024) and Ranković & Schwaller (2025), POST-GENERATION TS performs Thompson sampling over a fixed subset of candidates (of size 1000), here generated by the LLM prior to optimization. ACTOR CRITIC (Barto et al., 1983) maximizes $\mathbb{E}_{x \sim \pi^\theta}[\mu_x]$ rather than $\mathcal{V}$, and SOFT ACTOR CRITIC (Haarnoja et al., 2018) additionally adds entropy regularization, i.e.,

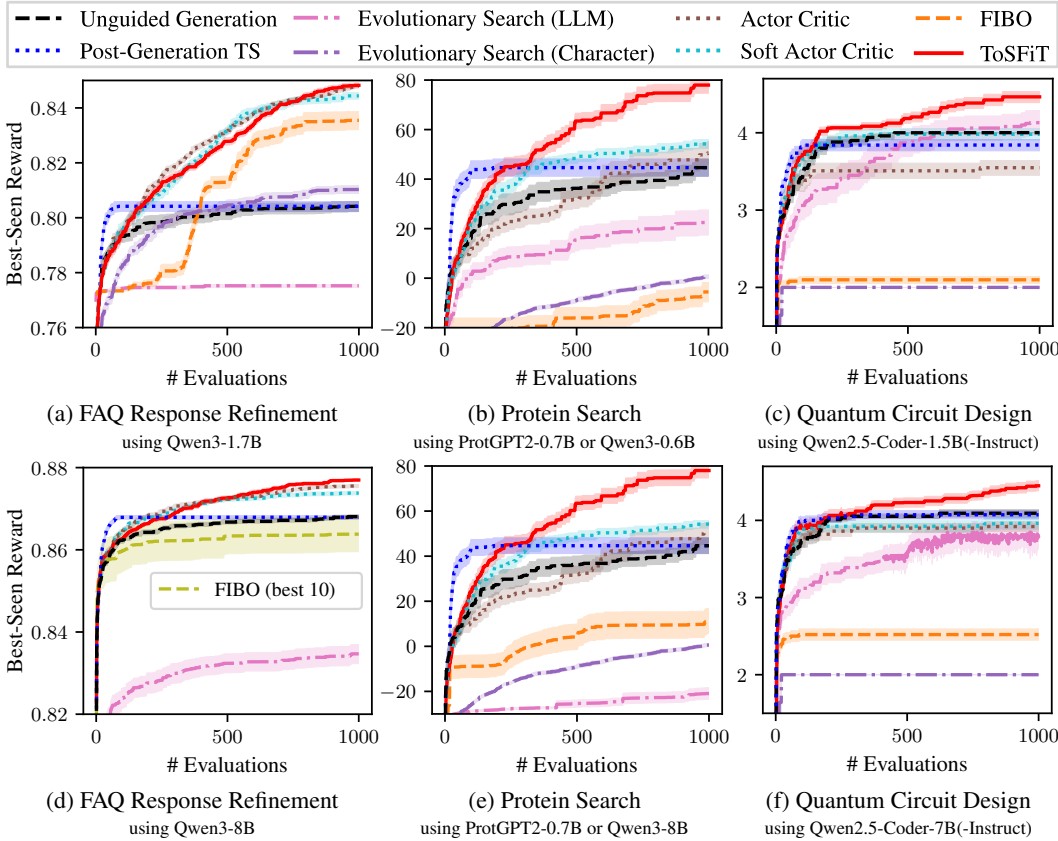

(a) FAQ Response Refinement
using Qwen3-1.7B

(b) Protein Search
using ProtGPT2-0.7B or Qwen3-0.6B

(c) Quantum Circuit Design
using Qwen2.5-Coder-1.5B(-Instruct)

(d) FAQ Response Refinement
using Qwen3-8B

(e) Protein Search
using ProtGPT2-0.7B or Qwen3-8B

(f) Quantum Circuit Design
using Qwen2.5-Coder-7B(-Instruct)

Figure 4: Across three tasks and within a collection of methods covering Bayesian optimization, reinforcement learning, as well as evolutionary search, TOSFIT exhibits state-of-the-art performance. In (b) and (e) EVOLUTIONARY SEARCH (LLM) and FIBO use Qwen for instruction following.

maximizes $\mathbb{E}_{x \sim \pi^\theta}[\mu_x - \alpha \ln \pi_x^\theta]$. EVOLUTIONARY SEARCH (CHARACTER) (Holland, 1975) treats solutions as sequences of text characters and applies biology inspired crossover and mutation operators. EVOLUTIONARY SEARCH (LLM) (Romera-Paredes et al., 2024) prompts a language model to conduct crossover and mutation of solutions. Finally, FIBO (Sutter et al., 2025) performs Thompson sampling fully in-context of an LLM by attending to a list of evaluated candidate solutions.

## 5.3 ToSFiT Obtains State-of-the-Art Sample Efficiency

Consider Figure 4. In all three tasks—FAQ Response Refinement, Protein Search, and Quantum Circuit Design—ToSFiT obtains state-of-the-art sample efficiency. Indeed, the best-seen reward of UNGUIDED GENERATION quickly saturates at a suboptimal level. Classical Bayesian optimization over a fixed subset of generated candidates (POST-GENERATION TS) identifies a good solution more efficiently, but remains confined to the initial sample pool. EVOLUTIONARY SEARCH as well as ACTOR CRITICS conduct undirected exploration without *optimism in the face of uncertainty*, i.e., an exploration bias guided by the reward potential of unexplored regions. As confirmed in Figure 5, the optimism of ToSFiT results in a more stable exploration-exploitation tradeoff than that of ACTOR CRITICS. FIBO is closest to ToSFiT in spirit, but relies on in-context learning instead of parameter updates. As can be seen from the result, this leads to significantly lower performance than ToSFiT. As FIBO keeps the generated candidate solutions as

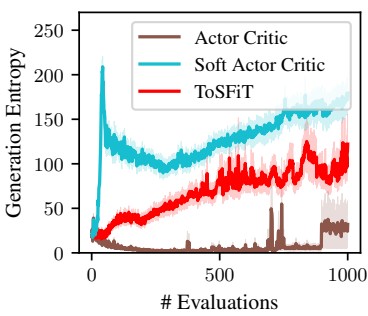

Figure 5: ToSFiT balances exploration with exploitation leading to stable policy diversity. Shown on Quantum Circuit Design, for other settings, see Section A.2.

generation context, it quickly runs out of memory and compute scales quadratically with the number of rounds, forcing us to compare against a top-10 truncated version of FIBO in Figure 4d.

Table 1: Among all LLM-based methods considered for optimization, only ToSFiT explores using optimism, conditions model generation for improved efficiency, and does not require the LLM to execute complex procedures via instruction following which sets requirements on the model size.

| | Exploits Promising Regions | Explores using Entropy | Explores using Optimism | Conditions Model Generation | Instruction Following Agnostic |
|---|---|---|---|---|---|
| UNGUIDED GENERATION | no | no | no | no | yes |
| POST-GENERATION TS | yes | yes | yes | no | yes |
| EVOLUTIONARY SEARCH | yes | no | no | yes | no |
| ACTOR CRITIC | yes | no | no | yes | yes |
| SOFT ACTOR CRITIC | yes | yes | no | yes | yes |
| FIBO | yes | yes | yes | yes | no |
| ToSFiT | yes | yes | yes | yes | yes |

**Scaling to Larger Models** Contrast the top row of experiments in Figure 4 with the bottom row (in Protein Search ProtGPT-2 is used twice for ToSFiT as well as all baselines without instruction following due to the absence of larger variants thereof). While increasing the model size is highly beneficial to FAQ Response Refinement, a natural language task, the tasks of Protein Search and Quantum Circuit Design profit far less. This matches the observations by Romera-Paredes et al. (2024) for code discovery. Indeed, novel discovery often requires leaving the training data manifold, i.e., the generative policy does not have to be tightly concentrated on the most promising solutions.

## 5.4 ToSFiT Remains Effective in Batched Bayesian Optimization

So far, we have focused on sequential Bayesian optimization, setting $b = 1$ in Algorithm 1. However, when observations are delayed or time-consuming, batched Bayesian optimization becomes preferable. Figure 6 considers a setting where up to 16 proteins can be synthesized and tested in parallel. While batching reduces sample efficiency—requiring more evaluations to reach a given reward—it improves iteration efficiency, achieving target performance in fewer rounds.

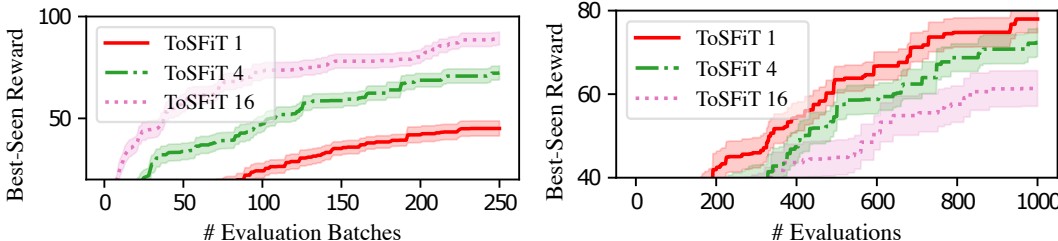

Figure 6: Protein Search with batched ToSFiT. The number to the right indicates the batch size $b$. Larger batches improve the iteration efficiency (left) with slightly lower sample efficiency (right).

## 5.5 ToSFiT Trades Off Computational Efficiency with Sample Efficiency

Consider Figure 7a, which explores the computational efficiency of ToSFiT. We do not plot the Actor Critics and Post-Generation TS, as their computational cost matches, respectively, ToSFiT and Unguided Generation with strictly worse reward. For $10'000$ rounds FIBO runs out of memory in its standard implementation, thus we once again implement FIBO attending only to the top-10 candidate solutions and their scores. This also decreases its compute costs drastically at the cost of limiting its sample efficiency. Note that the strong sample efficiency of ToSFiT compensates for the additional compute cost introduced by model fine-tuning, resulting in state-of-the-art computational efficiency. Next, consider Figure 7b. By investing additional compute through multiple steps of gradient ascent per round, the sample efficiency of ToSFiT can be further improved if needed.

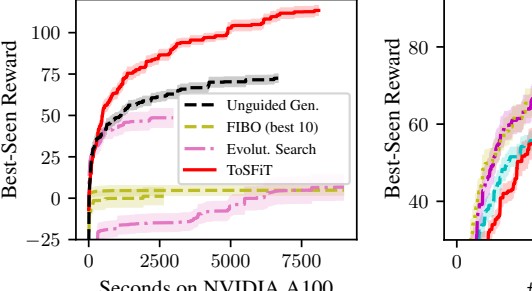

(a) Latency on fully batched ($B = b = 16$) Protein Search. FIBO and Evolutionary Search (LLM) are each tested with Qwen3-0.6B (fast) and Qwen3-8B (slow).

(b) The sample efficiency of ToSFiT on Protein Search is improved by increasing the number of steps of gradient ascent per round (denoted by $c$).

Figure 7: ToSFiT has state-of-the-art computational efficiency (a) and can trade off computational efficiency with sample efficiency (b).

## 5.6 ToSFiT Benefits from Strong Priors and Requires Careful Fine-Tuning

Lastly, we validate our theoretical insights. Theorem 1 highlights the importance of initializing the sampling policy with strong prior knowledge. Figure 8 provides empirical support: masking the number of qubits in the prompt (ToSFiT weak context) leads to a significantly worse reward than using the full prior context (ToSFiT). Note that uniform initialization fails to produce valid code. The theorem further suggests cautious adaptation to avoid drifting too far from the initialization. As shown in Figure 9, a large learning rate may initially improve performance by closely following VBOS, but eventually leads to forgetting the prior and stagnating as optimization becomes harder.

## 6 Related Work

As a remedy for the intractable maximization of acquisition functions over combinatorially large discrete domains, Bal et al. (2025) assume a factorization into the cartesian product of much smaller domains. They then identify local maxima as configurations where changing any of the factors individually decreases the acquisition function, and query the black-box function with the highest scoring game equilibrium. In contrast to ToSFiT, the domain must be structured to admit a factorization into cartesian products, and their optimization strategy does not leverage priors from LLMs.

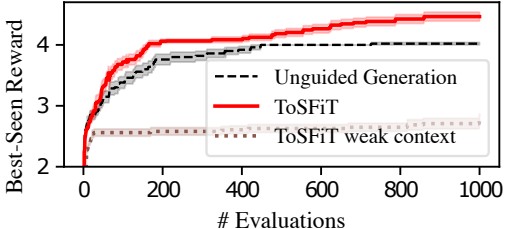 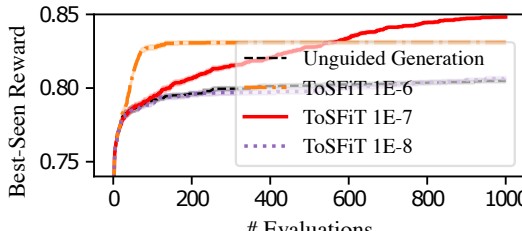

Figure 8: TOSFIT benefits markedly from a strong initial policy. On Quantum Circuit Design, using the most informative initial policy yields much better performance.

Figure 9: Careful adaptation to VBOS is essential to retain prior knowledge, here demonstrated for FAQ Refinement. The learning rate ($\eta$) is indicated to the right.

In a similar vein, Swersky et al. (2020) propose maximization of acquisition functions over discrete spaces through local mutations of a candidate solution. To conduct directed mutations, they learn a policy. In contrast to TOSFIT, their policy must be repeatedly applied to maximize the acquisition function, which quickly becomes expensive by itself, and shares similar limitations to Bal et al. (2025), such as the need for structured domains and lack of integration with pre-trained models.

There is extensive prior work on relaxing Bayesian optimization over discrete space to continuous space using variational autoencoders (Kusner et al., 2017; Lu et al., 2018; Griffiths & Hernández-Lobato, 2020; Notin et al., 2021; Lee et al., 2025). Acquisition function maximization is then tractably performed via gradient ascent in the continuous latent space. However, these methods require a task-specific pre-training phase for the autoencoder, whereas TOSFIT can be applied zero-shot through prompt-conditioning, thus benefiting from large-scale industrial LLM pretraining.

Finally, Ranković & Schwaller (2025) learn deep kernel features for Bayesian optimization with Gaussian processes. By treating the neural feature map as a hyperparameter, they use gradient ascent to maximize the marginal likelihood of observations during Bayesian optimization in a fully online fashion. Experimentally, they find much better optimization trajectories compared to fixed feature maps. That said, they do not address the intractability of acquisition function maximization. Their method is complementary to TOSFIT and could be integrated to learn task-specific embeddings.

## 7 CONCLUSION

We demonstrate that Thompson sampling can be efficiently scaled to large unstructured discrete domains by parameterizing the probability of maximality with a generative policy. Our theoretical analysis reveals that VBOS already leverages the kernel structure of Gaussian processes and achieves the strong regret bound of standard Thompson sampling and GP-UCB. Extending these results to approximate VBOS highlights the importance of initializing the sampling policy according to strong priors from pre-training and adapting it cautiously to maintain prior knowledge. These insights are supported by empirical results across three diverse tasks. Within a collection of methods covering Bayesian optimization, reinforcement learning, as well as evolutionary search, and to statistical significance, TOSFIT exhibits simultaneously state-of-the-art sample efficiency and computational efficiency. Together, our findings demonstrate the potential of combining foundation models with principled Bayesian optimization to tackle complex, discrete search problems.

### 7.1 LIMITATIONS AND FUTURE WORK

To ensure controlled comparisons, we evaluate TOSFIT under fixed feature maps—either derived from pre-trained embeddings or manually designed. A promising direction for future work is to learn deep, task-adaptive embeddings jointly with the Gaussian process, as in Ranković & Schwaller (2025), or replace the GP with more expressive reward models such as Bayesian neural networks or ensembles (Lakshminarayanan et al., 2017). Moreover, to reduce the computational and memory overhead introduced by fine-tuning, one could restrict updates to the last few layers of the generator. Finally, alternative strategies for optimizing the VBOS objective $\mathcal{V}(\pi)$—such as in-context conditioning—may offer a lightweight alternative to weight adaptation.

## ETHICS STATEMENT

This work does not involve human subjects, personally identifiable information, or sensitive data. All experiments were conducted using publicly available models. The proposed method, TOSFIT, is intended for research in optimization over discrete domains and is evaluated in controlled settings. We emphasize that TOSFIT is not intended for deployment in high-stakes decision-making without appropriate human oversight and domain-specific validation. As TOSFIT avoids the costly maximization of acquisition functions, environmental impact was a central consideration. No known conflicts of interest are associated with this work.

## REPRODUCIBILITY STATEMENT

Apart from the complete experimental code base provided publicly at `https://github.com/IBM/thompson-sampling-via-fine-tuning-of-llms`, pseudocode for TOSFIT is given in Algorithm 1. The baselines are also implemented in the code in a self-contained manner. The experimental environments for the three tasks considered is also provided in the code. Moreover, the provided README.md gives a step-by-step tutorial to reproduce all results in the paper. The hardware and software for running the experiments is described in detail in Section B of the Appendix. Furthermore, Section B.3 details the hyperparameter sweep conducted during development of this paper. In the evaluation, statistical significance is indicated via mean and standard error under 25 seeds, set to $0, \ldots, 24$. All theoretical statements are self-contained and include the complete set of assumptions. Both detailed proofs as well as proof sketches for the main results are provided in Section E of the Appendix. We distinguish between novel claims and prior statements by citing the appropriate works.

## ACKNOWLEDGMENT

This work is supported by the Swiss National Science Foundation (SNSF), grant 10002666.

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

# APPENDIX

## A    ADDITIONAL EXPERIMENTAL RESULTS

### A.1    COMPUTATIONAL COST OF TOSFIT

Bayesian optimization is designed to trade the number of evaluations of an expensive reward function against the computationally demanding computation of the reward posterior. In this spirit, we emphasize sample efficiency as the main consideration. Nevertheless, as measured in wall-clock time in Table 2, model fine-tuning only incurs a minor latency overhead (here $19\%$). As a result, despite more cost per round TOSFIT reaches higher reward scores at a fixed computational budget, i.e., it is computationally more efficient than all baselines (see Figure 7a).

Table 2: Wall-clock time per step of fully-batched TOSFIT ($b = B = 16$) on an A100 GPU for Protein Search. The majority of the runtime is spent on candidate generation.

| Autoregressive Generation | Model Fine-Tuning | GP Update | TOSFIT |
|---|---|---|---|
| $1.08 \pm 0.05$ s | $0.21 \pm 0.01$ s | $0.33 \pm 0.06$ ms | $1.29 \pm 0.05$ s |

Note that in the fully-batched setting considered here all generations are both used to estimate VBOS gradients and to provide observational feedback to the reward model. The 19% overhead observed in Table 2 stems from the additional non-auto-regressive backward pass during model fine-tuning. This computational overhead results in fewer generated candidates at a fixed budget.

### A.2    GENERATION DIVERSITY

A common failure point of reward-based fine-tuning is diversity collapse of the policy. As can be observed in Figure 10, ACTOR CRITIC (Barto et al., 1983) is strongly at risk. In contrast, SOFT ACTOR CRITIC (Haarnoja et al., 2018) retains diversity by directly regularizing the loss with an additional entropy term. Similarly, the objective of TOSFIT contains the additional term $\sqrt{-2 \ln \pi_x^\theta} \cdot \sigma_x$, which can be understood as entropy regularization targeted toward underexplored regions where $\sigma_x$ is large. As shown in Figure 10, this results in highly robust diversity trajectories. Moreover, in contrast to constant entropy regularization, such an uncertainty-aware exploration-exploitation tradeoff increases diversity in case of large uncertainty, and decreases diversity given sufficient observations.

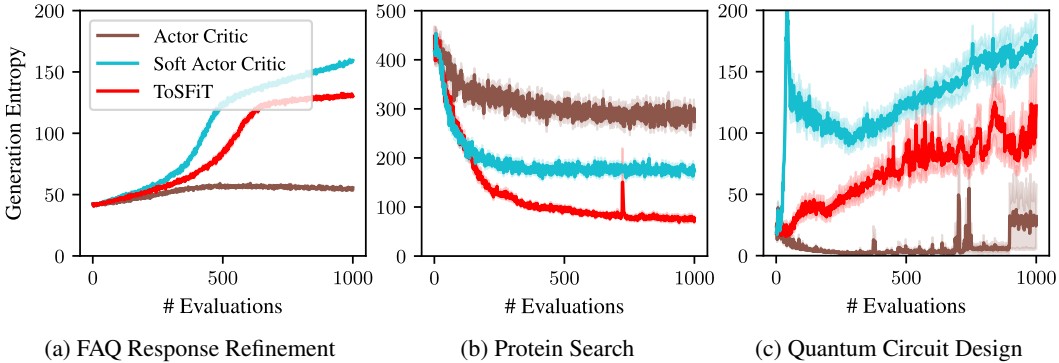

(a) FAQ Response Refinement      (b) Protein Search      (c) Quantum Circuit Design

Figure 10: TOSFIT directs exploration by incorporating uncertainty-aware entropy regularization via $\mathbb{E}_{x \sim \pi^\theta}[\sqrt{-2 \ln \pi_x^\theta} \cdot \sigma_x]$. It avoids the diversity collapse of ACTOR CRITIC and is more stable than SOFT ACTOR CRITIC (see Subfigure 10c). Moreover, by steering generation diversity with the reward uncertainty $\sigma_x$, TOSFIT enables both phases of exploration and phases of exploitation. Note that the behavior of each method yields distinct observation histories on top of loss discrepancies.

### A.3 LOSS AND RISK OF OVERFITTING

The loss function in ToSFiT is set to the negative VBOS objective, i.e., is a lower bound on $-\max_{x \in \mathcal{X}} R_x$. In contrast to training on a fixed dataset, the loss function evolves during optimization as more data is observed. Thus, the optimization dynamics typically do not lead to a monotonous increase in the objective, see Figure 11. Such non-stationarity leads to incorrect gradient statistics for optimizers like Adam (Kingma & Ba, 2015). Thus, we adopt vanilla stochastic gradient descent. Moreover, in contrast to a fixed dataset, there is no risk of overfitting the policy: the optimal policy is the one that maximizes the VBOS objective, as proven in Theorem 1. Indeed, both entropy regularization and uncertainty in the reward function are directly integrated into the loss. Overfitting of the reward surrogate $\mu_x \pm \sigma_x$ may still occur, but can be mitigated by adopting ensembles or Gaussian processes.

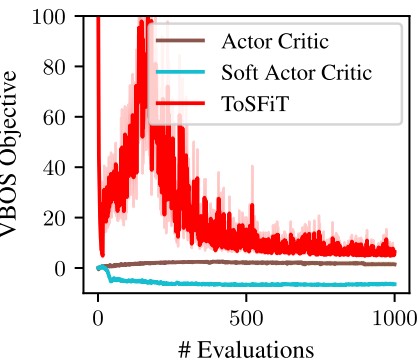

Figure 11: Evolution of the VBOS objective during Quantum Circuit Design using a Qwen2.5-Coder-1.5B.

## B EXPERIMENTAL DETAILS

To ensure reproducibility, we provide the setup for each of the experiments in Section 5. All experiments were conducted on compute nodes with an AMD EPYC 7763 64-Core CPU, 2TB RAM, and an NVIDIA A100-SXM4 GPU (80GB), running on Red Hat Enterprise Linux 9.4 with CUDA 12.4. Our Pytorch-based code (Paszke et al., 2019) is provided at `https://github.com/IBM/thompson-sampling-via-fine-tuning-of-llms`, including experimental configuration files and an environment file specifying the necessary packages and their versions.

### B.1 BASELINES

For **UNGUIDED GENERATION**, we simply generate and evaluate candidates according to the hyperparameters detailed in Section B.2.

For **POST-GENERATION TS**, we first generate a fixed candidate pool using the large language model with the same hyperparameters, where the pool size is set to the number of steps of Bayesian optimization. Then, we conduct standard Thompson sampling on the candidate subset using a Gaussian process surrogate model derived from the (deep) task-dependent embeddings.

For **ToSFiT**, we generate and evaluate candidates according to the same hyperparameters, but additionally follow Algorithm 1 to perform weight updates for consistency between the policy and the surrogate reward model, the same Gaussian process as in POST-GENERATION TS.

The **ACTOR CRITIC** baseline follows ToSFiT, but maximizes $\mathbb{E}_{x \sim \pi^\theta}[\mu_x]$ rather than $\mathcal{V}$.

**SOFT ACTOR CRITIC** additionally adds entropy regularization to ACTOR CRITIC, i.e., maximizes $\mathbb{E}_{x \sim \pi^\theta}[\mu_x - \alpha \ln \pi_x^\theta]$. The hyperparameter $\alpha$ is swept for each experimental setting.

For the **EVOLUTIONARY SEARCH** baselines, we generate new candidates by selecting two parents in each round independently via tournament selection with tournament size 3 and applying a crossover and mutation operation. We only let the 10 fittest members of the population survive. The initial population has one entry that acts as an example for the desired format. For the three tasks considered, the initial entry is set to

```
FAQ: How do I reset my password?
Q: How do I reset my password?
A: To reset your password, follow these steps:
Log in to your account (if you have access to the platform or
service you're using).
```

```
Locate the 'Forgot Password' or 'Reset Password' link, usually
found in the login form or menu.
Enter your email address or username associated with your account.
Follow the instructions sent to your email (or the confirmation
screen).
Complete the password reset process by entering a new password
and confirming it.
Submit the form to finalize the reset.
If you're unable to locate the "Forgot Password" option,
contact the support team for assistance.
```

```
MINDLLDISRIISGKMTLDRAEVNLTAIARQVVEEQRQAAEAKSIQLLCSTPDTNHYVFG
DFDRLKQTLWNLLSNAVKFTPSGGTVELELGYNAEGMEVYVKDSGIGIDPAFLPYVFDRF
RQSDAADSRNYGGLGLGLAIVKHLLDLHEGNVSAQSEGFGKGATFTVLLPLKPLKRELAA
VNRHTAVQQSAPLNDNLAGMKILIVEDRPDTNEMVSYILEEAGAIVETAESGAAALTSLK
SYSPDLVLSDIGMPMMDGYEMIEYIREWKTTKGG
```

```
qc.cx(0, 1)
```

**EVOLUTIONARY SEARCH (CHARACTER)** applies crossover by cutting each parent in half at a uniformly chosen location and stitching the pieces together. Mutation is performed through random substitution (at rate 0.05 per position) of the characters of the candidate solution with a letter from the vocabulary ACDEFGHIKLMNOPQRSTUVWY for Protein Search and one of the 95 printable ASCII characters for FAQ Response Refinement and Quantum Circuit Design. After substitution, random insertion & deletion of characters (at rate 0.01) is performed.

**EVOLUTIONARY SEARCH (LLM)** performs crossover and mutation in context through instruction following of the subsequent system prompt

```
You are conducting evolutionary search in context. You are
provided two candidate solutions. Propose a new distinct solution
by combining the candidate solutions and mutating the result.
Your novel candidate solution must be enclosed by
<candidate> </candidate>. Never repeat previous solutions. Your
search space is over FAQ responses to the question "How do I reset
my password?". /no_think
```

```
[...] Your search space is over amino acid sequences, with each
amino acid represented as a single-letter code./no_think
```

```
[...] Your search space is over 6-qubit Qiskit quantum circuits,
already prepared with `qc = QuantumCircuit(6)`.
```

Finally, **FIBO** implements Thompson sampling fully in context by attending to a list of evaluated candidate solutions, initially set to the same examples provided to EVOLUTIONARY SEARCH (LLM). The algorithm implements Thompson sampling by adopting the following system prompt:

```
You are conducting Bayesian optimization (Thompson sampling) fully
in context. You are provided a list of candidate solutions and the
rewards achieved by these solutions. Propose a new distinct
solution that maximizes the reward. Your novel candidate solution
must be enclosed by <candidate> </candidate>. Never repeat
previous solutions. Your search space is over FAQ responses to the
question "How do I reset my password?". /no_think
```

```
[...] Your search space is over amino acid sequences, with each
amino acid represented as a single-letter code./no_think
```

```
[...] Your search space is over 6-qubit Qiskit quantum circuits,
already prepared with `qc = QuantumCircuit(6)`.
```

## B.2 Experimental Details Shared Across Experiments

TOSFIT requires minimal task-specific hyperparameter tuning. Across all experiments with a Gaussian process reward model, a burn-in period of $m = 16$ steps is adopted and an exploration bonus of $4.0$ is applied to the prior amplitude inferred via MLM. The noise-to-amplitude ratio $\sigma_{nar}$ is always set to $0.01$, reflecting the absence of noise in the measurements, but ensuring stable inversion of the observation covariance matrix. Unless stated otherwise, the number of steps of gradient ascent per step of Bayesian optimization, $c$, is fixed at $1$ to maximize computational efficiency. To ensure stable estimation of VBOS gradients, the generation batch size $B$ is always set to $16$ with temperature set to $1.0$. We conduct unbatched Bayesian optimization ($b = 1$) unless otherwise specified. Learning rates are the only hyperparameters tuned per task: $1.0 \times 10^{-7}$ for FAQ refinement, $1.0 \times 10^{-5}$ for Protein Search, and $5.0 \times 10^{-6}$ for Quantum Circuit Design. The baseline SOFT ACTOR CRITIC requires an additional entropy regularization coefficient as hyperparameter and is swept across $\{0.01, 0.1, 1.0, 10.0\}$ for each experiment. We report on the best configuration.

**Implementation Details of TOSFIT**  To reduce the memory overhead of TOSFIT compared to UNGUIDED GENERATION (and in light of Section A.3), we do not adopt parameter-wise optimization statistics (Kingma & Ba, 2015). Instead, we use vanilla stochastic gradient descent with a fixed learning rate and without momentum. We do not store the gradients, but instead apply them directly during the backward pass. Also, while we perform fully-batched auto-regressive generation, the forward & backward passes to compute $\frac{d}{d\theta} \ln \pi_\theta(x_i)$ are performed in unbatched fashion with checkpointing. This crucially brings down memory consumption with little impact on runtime. All experiments were run on a single NVIDIA A100 GPU with 80 Gigabytes of VRAM.

## B.3 Choice of Hyperparameters

As mentioned above, TOSFIT requires minimal task-specific hyperparameter tuning. Indeed, only the learning rate is tuned for each task, based on one-dimensional grid search with search locations $\{5.0 \times 10^{-5}, 1.0 \times 10^{-5}, 5.0 \times 10^{-6}, 1.0 \times 10^{-6}, 5.0 \times 10^{-7}, 2.0 \times 10^{-7}, 1.0 \times 10^{-7}, 5.0 \times 10^{-8}\}$. Burn-in periods of $m \in \{4, 8, 16\}$ were tried out with $16$ being sufficiently large to ensure stable marginal likelihood maximization, yet small enough to not significantly impact sample efficiency due to delayed optimization. The exploration bonus was sweeped across the set $\{1.0, 2.0, 4.0, 8.0, 16.0\}$ on a toy task. A bonus of $4.0$ provided the best trade-off between exploration and exploitation, and generalized to strong performance on the reported experiments. The number of steps of gradient ascent per step of Bayesian optimization, $c$, was kept to $1$ in order to minimize the computational overhead compared to UNGUIDED GENERATION. For the generation batch size $B$, the values $\{4, 8, 16, 32, 64\}$ were tried out. $16$ was the smallest size where GPU utilization became acceptably large during auto-regressive generation. However, empirically $B$ did not significantly impact the simple regret trajectories during Bayesian optimization, suggesting that it could possibly be lowered for alternative models and/or hardware. The baseline SOFT ACTOR CRITIC requires an additional entropy regularization coefficient as hyperparameter and is swept across $\{0.01, 0.1, 1.0, 10.0\}$ for each experimental setup.

## B.4 FAQ Refinement

For TOSFIT, the ACTOR CRITICS, UNGUIDED GENERATION, and POST-GENERATION TS, we use the Qwen3-1.7B/Qwen3-8B model with the system prompt *You are a helpful assistant.* followed by the prompt *Write an FAQ response to the question "How do I reset my password?". /no_think*. We adopt a learning rate of 1.0E-7/2.0E-7. As kernel features, we extract the first 256 entries of the Qwen3-Embedding-0.6B model (Zhang et al., 2025) with input set to the concatenation of prompt and response. We normalize the result to the unit sphere and add an additional constant 1 entry to act as a bias. The reward is modeled as the alignment to the unknown ground-truth response *To reset your password, go to the login page and click the "Forgot Password" link. Enter your registered email address, and we'll send you instructions to create a new password. Make sure to check your spam folder if you don't see the email within a few minutes.*, computed through the cosine similarity between the full Qwen3-Embedding-0.6B embeddings of the candidate and the ground-truth.

### B.5 PROTEIN SEARCH

For TOSFIT, the ACTOR CRITICS, UNGUIDED GENERATION, and POST-GENERATION TS, we use the ProtGPT2 model (Ferruz et al., 2022) with initial token <|*endoftext*|> and a learning rate of 1.0E-5. As kernel features, we normalize each 1280 dimensional token embedding of the generated sequence to the unit sphere, average across the sequence length, normalize the result to the unit sphere, and add an additional constant 1 entry to act as a bias. The reward is set to the negative thermal instability index (Guruprasad et al., 1990) implemented in the BioPython library (Cock et al., 2009).

### B.6 QUANTUM CIRCUIT DESIGN

For TOSFIT, the ACTOR CRITICS, UNGUIDED GENERATION, and POST-GENERATION TS, we use the Qwen2.5-Coder-1.5B/Qwen2.5-Coder-7B model (Hui et al., 2024) prompted with fill-in-the-middle (Bavarian et al., 2022) to generate a Qiskit circuit (Javadi-Abhari et al., 2024):

```
<|fim_prefix|>
def circuit():
    qc = QuantumCircuit(6)
<|fim_suffix|>
    return qc
qc = circuit()
state = Statevector.from_instruction(qc)
<|fim_middle|>}.
```

We adopt a learning rate of 5.0E-6/1.0E-5. As kernel features, we take all 210 two-qubit Pauli observables, which are efficiently simulated by quantum computers, and concatenate an additional code invalidity bit. If the code invalidity bit is set to 1, all other features are set to zero. Finally, we add a constant 1 entry to act as a bias. The reward is set to the negative energy of the prepared state according to an unknown Hamiltonian. The Hamiltonian is constructed through the following weighted linear combination of Pauli terms:

```
(ZIIIIII, -1.0)
(IZIIIII, 1.0)
(IIXIIII, -1.0)
(IIIYIII, -1.0)
(IIIIZZI, -1.0)
(IIIIXXI, -1.0)
(IZZIIII, -1.0)
(YIIIIIX, -0.5)
(IIZIIIX, -0.5).
```

Note that the Hamiltonian heavily features superposition-inducing $X$ and $Y$ operators and pair-wise interaction terms, necessitating entanglement to reach the lowest energy state.

## C STANDARDIZED RLOO IS GRPO

TOSFIT crucially relies on the score trick (Williams, 1992) for unbiased gradient estimation of the VBOS objective (see Proposition 1). The generalized score trick states that for $x_i \overset{i.i.d.}{\sim} \pi^\theta$

$$\frac{d}{d\theta}\mathbb{E}_{x\sim\pi^\theta}[r(x,\theta)] = \mathbb{E}_{x\sim\pi^\theta}\left[\left(\hat{r}(x,\theta)+\xi\right)\frac{d}{d\theta}\ln\pi^\theta(x)\right] \approx \frac{1}{B}\sum_i(\hat{r}(x_i,\theta)+\xi_i)\frac{d}{d\theta}\ln\pi^\theta(x_i),$$

where $\hat{r}(x,\theta) = \frac{d}{d\theta}r(x,\theta)/\frac{d}{d\theta}\ln\pi^\theta(x) + r(x,\theta)$. Note that for unbiased estimation the free baseline $\xi \in \mathbb{R}$ can be chosen differently for each $i \in \{1,\ldots,B\}$ *as long as it does not depend on* $x_i$. One effective baseline that greatly reduces the variance is to set $\xi_i = \frac{1}{B-1}\sum_{j\neq i}\hat{r}(x_j,\theta)$ (Kool et al., 2019), resulting in the 'Reinforce Leave One Out' (RLOO) advantage function $\hat{r}_j - \frac{1}{B-1}\sum_{j\neq i}\hat{r}(x_j,\theta)$. As we show in Proposition 3, standardization of RLOO recovers the advantage function introduced by Shao et al. (2024) for Group Relative Policy Optimization.

**Proposition 3.** *Consider the 'Reinforce Leave One Out' (Kool et al., 2019) advantage function given by $\hat{r}_i - \frac{1}{B-1}\sum_{j\neq i}\hat{r}_j$ for i.i.d. samples $\hat{r}_j$ with $j \in \{1,\ldots,B\}$. Then the expected advantage is zero and an unbiased estimator of the variance is $\frac{1}{B}\sum_h(\hat{r}_h - \frac{1}{B-1}\sum_{l\neq h}\hat{r}_l)^2$. Moreover, standardizing RLOO results precisely in the advantage function employed by Shao et al. (2024) for 'Group Relative Policy Optimization' (GRPO):*

$$\underbrace{\frac{\hat{r}_i - \frac{1}{B-1}\sum_{j\neq i}\hat{r}_j}{\sqrt{\frac{1}{B}\sum_h(\hat{r}_h - \frac{1}{B-1}\sum_{l\neq h}\hat{r}_l)^2}}}_{\text{standardized RLOO}} = \underbrace{\frac{\hat{r}_i - \frac{1}{B}\sum_j\hat{r}_j}{\sqrt{\frac{1}{B}\sum_h(\hat{r}_h - \frac{1}{B}\sum_l\hat{r}_l)^2}}}_{\text{GRPO}}$$

# D    CONSTANT-TIME GP INFERENCE

To ensure efficient runtime in large-scale settings and to integrate with deep neural features $\phi : X \to \mathbb{R}^d$, we consider Gaussian processes (GPs) with linear kernels over $\mathbb{R}^d$ (Rasmussen & Williams, 2006). As we explain in the following, our implementation requires a constant $\Theta(d^2)$ memory and compute for conditioning on an observation, inferring posterior mean & variance at a point, and even conducting marginal likelihood maximization. In contrast to a straightforward implementation, it does not increase with the number of past observations.

## D.1    SETTING

Let $R \sim \mathcal{GP}(\nu, \lambda^2\bar{k})$ with linear kernel $\bar{k}_{x,z} = \phi(x)^T\phi(z)$ for $\phi : X \to \mathbb{R}^d$. Here, $\lambda \in \mathbb{R}_+$ and $\nu \in \mathbb{R}$ denote parameters of the prior. We assume that observations $y_x$ of $r_x$ come with i.i.d. additive Gaussian noise, i.e., $Y_x \sim R_x + \lambda\varepsilon$ for $\varepsilon \sim \mathcal{N}(0, \sigma_{noise}^2)$ independent across observations. Then, given a batch of observations $(x_O, y_O)$, the posterior GP takes the closed form

$$\mu_x = \nu + \bar{k}_{x,x_O}^T(\bar{k}_{x_O,x_O} + \sigma_{nar}^2 I_s)^{-1}(y_O - \nu), \tag{3}$$

$$k_{x,z}/\lambda^2 = \bar{k}_{x,z} - \bar{k}_{x,x_O}^T\underbrace{(\bar{k}_{x_O,x_O} + \sigma_{nar}^2 I_s)^{-1}}_{\Sigma = \Sigma_{Y_O}/\lambda^2}\bar{k}_{z,x_O}, \tag{4}$$

where $\sigma_{nar} = \sigma_{noise}/\lambda$ denotes the noise to amplitude ratio and $\Sigma_{Y_O} = \lambda^2\Sigma$ the observation covariance matrix.

## D.2    CLOSED-FORM MLM

In our setup, the only free parameters are the global scale $\lambda \in \mathbb{R}_+$ and the global offset $\nu \in \mathbb{R}$. We avoid optimizing $\sigma_{nar} \in \mathbb{R}_+$ due to (1) its dual use for numerical stability of the inverse of $\Sigma$ and (2) analytical intractability of general marginal likelihood maximization. As we prove in Proposition 4, in this setting the marginal data likelihood can be maximized efficiently in closed form.

**Proposition 4.** *Let $Y \sim \mathcal{N}(\nu, \lambda^2\Sigma)$. Then for $y \in \mathbb{R}^s$, the (marginal) likelihood $p(Y = y)$ is maximized if*

$$\nu^{opt} = \frac{y^T\Sigma^{-1}\mathbb{1}}{\mathbb{1}^T\Sigma^{-1}\mathbb{1}} \text{ and } \lambda^{opt} = \sqrt{\frac{(y-\nu^{opt}\mathbb{1})^T\Sigma^{-1}(y-\nu^{opt}\mathbb{1})}{s}}.$$

## D.3    BYPASSING THE COVARIANCE MATRIX

The conditioning on observations in Equations (3) and (4) as well as the closed-form marginal likelihood maximization outlined in Proposition 4 both rely on computing, storing, and inverting the unscaled prior data covariance matrix $\Sigma$. Since $\Sigma$ grows in $s$, trivial application of the closed formulas would prevent constant-time GP inference. However, for linear kernels, a computationally much more efficient form can be exploited instead. Define $\Phi := [\phi(x_1) \quad \ldots \quad \phi(x_s)] \in \mathbb{R}^{d\times s}$.

Then, per the Sherman-Morrison-Woodbury formula

$$\Sigma^{-1} = (\Phi^T\Phi + \sigma_{nar}^2 I_s)^{-1} = \frac{1}{\sigma_{nar}^2}(I_s - \Phi^T(\Phi\Phi^T + \sigma_{nar}^2 I_d)^{-1}\Phi),$$

$$\Phi\Sigma^{-1} = \underbrace{(\Phi\Phi^T + \sigma_{nar}^2 I_d)}_{=:\Psi}{}^{-1}\Phi, \text{ and} \tag{5}$$

$$I_d - \Phi\Sigma^{-1}\Phi^T = I_d - \Psi^{-1}\Phi\Phi^T = \sigma_{nar}^2\Psi^{-1}. \tag{6}$$

Note that the expression to the left of Equation (5) underlies the conditioning of the mean in Equation (3) and that the left-hand expression of Equation (6) is used to condition the kernel in Equation (4). Therefore, given $\Psi^{-1} \in \mathbb{R}^{d\times d}$, and $\Phi y_O, \Phi\mathbb{1} \in \mathbb{R}^d$, computing $\mu_x$ and $k_{x,z}$ according to Equations (3) and (4) can be performed in $\Theta(d^2)$ steps for any prior parameter-pair $(\nu, \lambda)$. Furthermore, having additionally access to $y_O^T y_O, \mathbb{1}^T\mathbb{1}, y_O^T\mathbb{1} \in \mathbb{R}^n$ allows performing closed-from marginal likelihood maximization according to Proposition 4, also in $\Theta(d^2)$ steps. So, to avoid scaling in the number of past observations $s$, we only need to keep track of $\Psi^{-1}, \Phi y_O, \Phi\mathbb{1}, y_O^T y_O, \mathbb{1}^T\mathbb{1}$ and $y_O^T\mathbb{1}$ during Bayesian optimization, i.e., while we add columns to $\Phi$ and new entries to $y_O$ and $\mathbb{1}$. For most of these terms, this is trivial. Only $\Psi^{-1}$ requires again relying on Sherman-Morrison-Woodbury, i.e.,

$$\begin{aligned}\Psi_{new}^{-1} &= (\Phi\Phi^T + \sigma_{nar}^2 I + \phi(x)\phi(x)^T)^{-1}\\ &= \Psi^{-1} - \Psi^{-1}\phi(x)\frac{1}{1+\phi(x)^T\Psi^{-1}\phi(x)}\phi(x)^T\Psi^{-1}.\end{aligned}$$

Keeping track of these quantities requires $\Theta(d^2)$ memory with each additional observation demanding $\Theta(d^2)$ compute, validating our complexity claims.

# E  PROOFS

Before diving into the individual proofs of the theorems and propositions introduced in the paper, we take the opportunity to present proof sketches for the main results.

## E.1  PROOF SKETCHES

**Proposition 1**  We show concavity by deriving the Hessian of the VBOS objective, which is a diagonal matrix with non-positive entries. To derive the unbiased estimator of the gradients, we apply the generalized score trick for reward functions that depend on the model parameters $\theta$.

**Proposition 2**  After verifying convexity of $f$, we simply expand the definition of the Bregman divergence and use that $\nabla f(\tilde{\pi}) = \mu - c\mathbb{1}$ for a Lagrange multiplier $c$, since $\tilde{\pi}$ is known to maximize the VBOS objective on the probability simplex and known to lie in the relative interior.

**Theorem 1**  We first use linearity of expectation and the law of total expectation to individually consider the expected instantaneous regret $\mathbb{E}[\mathbb{E}[R^* - R_{x^t}|\mathcal{H}^t]]$ at each time step. Next, we leverage the technical result from Corollary 2, established by prior work, to upper bound $\mathbb{E}[R^*|\mathcal{H}^t]$ with $\mathcal{V}(\tilde{\pi}^t|\mathcal{H}^t)$. In contrast to previous analysis by Tarbouriech et al. (2024), we incorporate practical limitations of gradient-based objective maximization by replacing $\mathcal{V}(\tilde{\pi}^t)$ with $D_{\sigma^t}(\pi^t, \tilde{\pi}^t) + \mathcal{V}(\pi^t)$. Since $x^t$ follows $\pi^t$ in practice, and not $\tilde{\pi}^t$, it is essential to have $\mathcal{V}(\pi^t)$ instead of $\mathcal{V}(\tilde{\pi}^t)$ as an upper bound. Now, following previous analysis, we simplify $\mathcal{V}(\pi^t|\mathcal{H}^t) - \mathbb{E}[R_{x^t}|\mathcal{H}^t]$ to $\mathbb{E}[\sigma_{x^t}\sqrt{-2\ln\pi^t(x^t)}|\mathcal{H}^t]$ and apply the Cauchy-Schwarz inequality to arrive at the product of a term that scales in $\sqrt{T}$ and $\sqrt{\sum_t \mathbb{E}[\sigma_{x^t}^2]}$. Whereas Tarbouriech et al. (2024) bound the latter based on the pigeon-hole principle, we instead bound it by the maximal information gain $\gamma^T$, a technique introduced by Srinivas et al. (2010) to analyze multi-armed bandits with correlated rewards.

## E.2 FORMAL PROOFS

**Proposition 1.** *Consider the* VBOS *objective* $\mathcal{V}(\pi^\theta) := \mathbb{E}_{x \sim \pi^\theta}[\mu_x + \sqrt{2 \ln(1/\pi_x^\theta)} \cdot \sigma_x]$. *The* VBOS *objective* $\mathcal{V}$ *is concave (strictly if* $\sigma_x > 0 \; \forall x$*) and its gradients are*

$$\frac{d}{d\theta}\mathcal{V}(\pi^\theta) = \mathbb{E}_{x \sim \pi^\theta}\Big[\big(\mu_x \underbrace{-\xi - v^{\text{-}1}(\pi_x^\theta) \cdot \sigma_x}_{-\mu_x^\theta \text{ for } \xi = \kappa}\big) \cdot \frac{d \ln \pi_x^\theta}{d\theta}\Big].$$

$\xi \in \mathbb{R}$ *is an arbitrary baseline and* $\text{-}v^{\text{-}1}(u) = \sqrt{\text{-}2 \ln(u)} - 1/\sqrt{\text{-}2 \ln(u)} \sim \sqrt{\text{-}2 \ln(u)}$ *as* $u \to 0$.

*Proof of Proposition 1.* To show concavity, we follow the proof of Proposition 4 in Menet et al. (2025) by noting that $v' > 0$ and considering the (diagonal) Hessian of $\mathcal{V}$, where

$$\frac{\partial}{\partial r_x}\mathcal{V}(r) = \mu_x + \sigma_x\Big(\sqrt{2 \ln(1/r_x)} - \frac{1}{\sqrt{2 \ln(1/r_x)}}\Big) = \mu_x - \sigma_x v^{\text{-}1}(r_x), \text{ and}$$

$$\frac{\partial^2}{\partial r_x \partial r_z}\mathcal{V}(r) = -\sigma_x \mathbb{1}_{x=z}\frac{1}{v'(v^{\text{-}1}(r_x))}\begin{cases} \leq 0 & x = z \\ = 0 & x \neq z \end{cases}.$$

Note the use of the inverse function rule $\frac{d}{da}v^{\text{-}1}(a) = 1/\frac{d}{db}v(b)|_{b=v^{\text{-}1}(a)}$. In case $\sigma_x > 0 \; \forall x$, strict concavity is even ensured. So, we only need to show the formula for gradients. We make use of the score trick, which states that

$$\frac{d}{d\theta}\mathbb{E}_{x \sim \pi^\theta}[f(x, \pi^\theta(x))] = \mathbb{E}_{x \sim \pi^\theta}\Big[\big(f(x,y) + y\tfrac{d}{dy}f(x,y)\big)\big|_{y=\pi^\theta(x)}\frac{d \ln \pi^\theta(x)}{d\theta}\Big].$$

Plugging in and using that $\mathbb{E}_{x \sim \pi^\theta}[\xi\frac{d \ln \pi_x^\theta}{d\theta}] = 0$ for any constant $\xi \in \mathbb{R}$ (that neither depends on $\theta$ nor $x$) results in the desired expression:

$$\begin{aligned}
\frac{d}{d\theta}\mathcal{V}(\pi^\theta) &= \mathbb{E}_{x \sim \pi^\theta}\Big[\big(\mu_x + \sqrt{\text{-}2\ln(\pi_x^\theta)} \cdot \sigma_x - \frac{1}{\sqrt{\text{-}2\ln(\pi_x^\theta)}} \cdot \sigma_x\big)\frac{d \ln \pi_x^\theta}{d\theta}\Big] \\
&= \mathbb{E}_{x \sim \pi^\theta}\Big[\big(\mu_x - v^{\text{-}1}(\pi_x^\theta) \cdot \sigma_x\big)\frac{d \ln \pi_x^\theta}{d\theta}\Big] \\
&= \mathbb{E}_{x \sim \pi^\theta}\Big[\big(\mu_x - \xi - v^{\text{-}1}(\pi_x^\theta) \cdot \sigma_x\big)\frac{d \ln \pi_x^\theta}{d\theta}\Big].
\end{aligned}$$

$\square$

**Proposition 2.** *Let* $\sigma \in \mathbb{R}_+^{|X|}$. *For the convex* $f(p) := -\sum_x p_x \sigma_x \sqrt{-2 \ln p_x}$, *define the Bregman divergence* $D_\sigma(p, q) = f(p) - f(q) - \langle \nabla f(q), p - q \rangle$. *Then the Bregman divergence of any* $\pi \in \Delta^{|X|-1}$ *from the maximizer* $\tilde{\pi} := \arg\max_{p \in \Delta^{|X|-1}} \mathcal{V}(p)$ *is given by* $D_\sigma(\pi, \tilde{\pi}) = \mathcal{V}(\tilde{\pi}) - \mathcal{V}(\pi)$.

*Proof of Proposition 2.* We follow parts of the proof of Lemma 5 by Tarbouriech et al. (2024). First, recall from Equation (2) the closed-form expression for $\tilde{\pi} = v(\frac{\mu_x - \kappa^*}{\sigma_x})$, where $v$ is a cumulative distribution function with $v^{\text{-}1}(u) = 1/\sqrt{-2 \ln u} - \sqrt{-2 \ln u}$. Next, notice that due to $v$ being a cumulative distribution function one has $-\frac{d^2}{dp_x^2}p_x\sqrt{-2 \ln p_x} = \frac{d}{dp_x}v^{\text{-}1}(p_x) = 1/(v'(v^{\text{-}1}(p_x))) > 0$. As such, $f(p)$ is continuously-differentiable and strictly convex (positive definite Hessian), i.e., the Bregman divergence is well-defined. Next, recall that $\mathcal{V}(p) = \sum_x p_x \mu_x + \sigma_x p_x \sqrt{-2 \ln p_x}$, and thus $\mathcal{V}(p) = \langle \mu, p \rangle - f(p)$. Since $\tilde{\pi}$ maximizes $\mathcal{V}$ and is in the relative interior of the simplex (the objective is concave and at the border its derivatives blow up), the Karush-Kuhn-Tucker conditions give $c\mathbb{1} = \nabla\mathcal{V}(\tilde{\pi}) = \mu - \nabla f(\tilde{\pi})$ for some $c \in \mathbb{R}$, i.e., $\nabla f(\tilde{\pi}) = \mu - c\mathbb{1}$ and thus $\mathcal{V}(\pi) = \langle \nabla f(\tilde{\pi}), \pi \rangle - f(\pi) + c$. Plugging into $\mathcal{V}(\tilde{\pi}) - \mathcal{V}(\pi)$ then gives precisely the definition of $D_\sigma(\pi, \tilde{\pi})$.

$\square$

**Theorem 1.** *Let* $R \sim \mathcal{N}(\mu, K)$ *with* $K_{x,x} \leq 1 \; \forall x \in X$ *and additive observation noise* $\mathcal{N}(0, \sigma_n^2)$.[4] *If* $R$ *is observed at* $x^t \sim \pi^t$ *for a policy* $\pi^t$ *depending on history* $\mathcal{H}^t$, *then*

$$\mathbb{E}[\textstyle\sum_{t=1}^T R^* - R_{x^t}] \leq \sqrt{C_{\sigma_n}HT\gamma^T} + \mathbb{E}\textstyle\sum_{t=1}^T D_{\sigma^t}(\pi^t, \tilde{\pi}^t).$$

$C_{\sigma_n} := 4/\ln(1 + \sigma_n^{-2})$ *is a constant,* $H := \frac{1}{T}\sum_t H[\pi^t|\mathcal{H}^t]$ *is the expected average entropy of the policy and hence upper bounded by* $\ln|X|$, $\gamma^T := \max_{L^T} I(Y_{L^T}; R)$ *is the maximum information gain for* $T$ *observation locations* $L^T$, *and* $\tilde{\pi}^t$ *is the unconstrained maximizer of* VBOS *given* $\mathcal{H}^t$.

---

[4]The theorem also holds for heteroscedastic additive Gaussian noise by replacing $\sigma_n$ with $\max_{x \in X} \sigma_n(x)$.

*Proof of Theorem 1.* First, note that if $R$ is a multivariate Gaussian, then for $\mathcal{H}^t$ the history of observations at step $t$, $R^t := R|\mathcal{H}^t$, is also a multivariate Gaussian $\forall t$. Next, we leverage Corollary 2 to upper bound the regret

$$
\begin{aligned}
\mathbb{E}[\textstyle\sum_t R^* - R_{x^t}] &= \textstyle\sum_t \mathbb{E}[\mathbb{E}[R^* - R_{x^t}|\mathcal{H}^t]] \\
&\leq \textstyle\sum_t \mathbb{E}[\mathcal{V}(\tilde{\pi}^t|\mathcal{H}^t) - \mathbb{E}_{x^t \sim \pi^t}[\mu_{x^t}|\mathcal{H}^t]] \\
&= \textstyle\sum_t \mathbb{E}[\mathcal{V}(\pi^t|\mathcal{H}^t) + D_{\sigma^t}(\pi^t, \tilde{\pi}^t) - \mathbb{E}_{x^t \sim \pi^t}[\mu_{x^t}|\mathcal{H}^t]] \\
&= \textstyle\sum_t \mathbb{E}[\mathbb{E}_{x^t \sim \pi^t}[\sigma_{x^t}\sqrt{-2\ln \pi^t(x^t)}|\mathcal{H}^t] + D_{\sigma^t}(\pi^t, \tilde{\pi}^t)].
\end{aligned}
$$

Note the application of Proposition 2, which allows us to bound the cumulative regret even if the policy $\pi^t$ differs from the one suggested by VBOS, i.e., $\tilde{\pi}^t$. Next, we use the Cauchy-Schwarz inequality as well as Lemma 4 to uncover the dependency on maximum information gain, a measure of the complexity of the kernel. We get

$$
\begin{aligned}
\textstyle\sum_t \mathbb{E}[\mathbb{E}_{x^t \sim \pi^t}[\sigma_{x^t}\sqrt{-2\ln \pi^t(x^t)}|\mathcal{H}^t]] &\leq \sqrt{\textstyle\sum_t \mathbb{E}[\mathbb{E}_{x^t \sim \pi^t}[-\ln(\pi^t(x^t))|\mathcal{H}^t]]}\sqrt{\textstyle\sum_t \mathbb{E}[\mathbb{E}[2\sigma_{x^t}^2|\mathcal{H}^t]]} \\
&\leq \sqrt{\textstyle\sum_t H[\pi^t|\mathcal{H}^t]}\sqrt{C_{\sigma_n}\gamma^T}.
\end{aligned}
$$

$\square$

**Lemma 1** (Donsker-Varadhan variational representation). *Fix two probability distributions $p : \Sigma \to [0,1]$ and $q : \Sigma \to [0,1]$ over the measurable space $(\Omega, \Sigma)$ such that $p$ is absolutely continuous with respect to $q$ ($p \ll q$). Then*

$$
D_{KL}[p||q] = \sup_X \{\mathbb{E}_p[X] - \ln \mathbb{E}_q[\exp X]\},
$$

*where the supremum is taken over all measurable $X : \Omega \to \mathbb{R}$ such that $\mathbb{E}_p[X]$ and $\mathbb{E}_q[\exp X]$ are well-defined.*

*Proof.* The provided proof is a generalization of Theorem 3.2 by Gray (2011) from discrete spaces to arbitrary probability spaces. Since $p \ll q$, there exists a Radon-Nykodym derivative $\frac{dp}{dq}(\omega)$, i.e., it holds $p(\mathcal{A}) = \int_{\mathcal{A}} \frac{dp}{dq}(\omega)dq(\omega)$ for a function $\frac{dp}{dq}(\omega)$ uniquely defined up to a set of $q$-measure zero. Now, let $X : \Omega \to \mathbb{R}$ be any random variable such that $\mathbb{E}_p[X]$ and $\mathbb{E}_q[\exp X]$ are well-defined. Then

$$
\begin{aligned}
D_{KL}[p||q] - (\mathbb{E}_p[X] - \ln(\mathbb{E}_q[\exp X])) &= \mathbb{E}_p[\ln \frac{dp}{dq}(\omega)] - \mathbb{E}_p[\ln \frac{\exp X}{\mathbb{E}_q[\exp X]}] \\
&= \mathbb{E}_p[\ln \left(\frac{dp}{dq}(\omega)\frac{\mathbb{E}_q[\exp X]}{\exp(X)}\right)] \\
&= \mathbb{E}_p[\ln \frac{dp}{d\lambda}] = D_{KL}[p||\lambda] \geq 0,
\end{aligned}
$$

where we defined the probability measure $\lambda(\mathcal{A}) = \int_{\mathcal{A}} \exp(X(\omega))/\mathbb{E}_q[\exp X]dq(\omega)$, substituted $\frac{\mathbb{E}_q[\exp X]}{\exp(X)} = 1/\frac{d\lambda}{dq} = \frac{dq}{d\lambda}$, and simplified the derivatives. $\square$

**Lemma 2** (Conditioned KL-divergence). *Consider the probability space $(\Omega, \Sigma, \mathbb{P})$ and an event $\mathcal{B} \in \Sigma$ of non-zero probability, i.e. $\mathbb{P}[\mathcal{B}] > 0$. Then*

$$
D_{KL}[\mathbb{P}[\,\cdot\,|\mathcal{B}] \,||\, \mathbb{P}] = -\ln \mathbb{P}[\mathcal{B}].
$$

*Proof.* The proof follows Menet et al. (2025). According to the definition of conditional expectation it holds

$$
\mathbb{P}[\mathcal{A}\,|\mathcal{B}] = \frac{\mathbb{P}[\mathcal{A} \cap \mathcal{B}]}{\mathbb{P}[\mathcal{B}]} = \int_{\mathcal{A}} \frac{\mathbb{1}_{\omega \in \mathcal{B}}}{\mathbb{P}[\mathcal{B}]}\,d\mathbb{P}(\omega) \qquad \forall \mathcal{A} \in \Sigma,
$$

where we recognize absolute continuity $\mathbb{P}[\,\cdot\,|\mathcal{B}] \ll \mathbb{P}$ and identify the Radon-Nykodym derivative $\frac{d\mathbb{P}[\,\cdot\,|\mathcal{B}]}{d\mathbb{P}}(\omega) = \frac{\mathbb{1}_{\omega \in \mathcal{B}}}{\mathbb{P}[\mathcal{B}]}$. Plugging the derivative into the definition of Kullback-Leibler divergence

results in the desired term:

$$
\begin{aligned}
D_{KL}[\mathbb{P}[\,\cdot\,|\mathcal{B}] \,||\, \mathbb{P}] &:= \int_{\Omega} \ln \frac{d\mathbb{P}[\,\cdot\,|\mathcal{B}]}{d\mathbb{P}} d\mathbb{P}[\,\cdot\,|\mathcal{B}] \\
&= \int_{\Omega} \ln \Big(\frac{d\mathbb{P}[\,\cdot\,|\mathcal{B}]}{d\mathbb{P}}\Big)\frac{d\mathbb{P}[\,\cdot\,|\mathcal{B}]}{d\mathbb{P}} d\mathbb{P} \\
&= \int_{\Omega} \ln \Big(\frac{\mathbb{1}_{\omega \in \mathcal{B}}}{\mathbb{P}[\mathcal{B}]}\Big)\frac{\mathbb{1}_{\omega \in \mathcal{B}}}{\mathbb{P}[\mathcal{B}]} d\mathbb{P} \\
&= -\ln \mathbb{P}[\mathcal{B}].
\end{aligned}
$$

$\square$

**Definition 1.** *The (Cramér) rate function $\Lambda^*$ of a random variable $X : \Omega \to \mathbb{R}$ is defined as the convex conjugate of the cumulant generating function $\Lambda$, i.e.,*

$$
\Lambda^*(\alpha) := \sup_{\beta > 0} \alpha\beta - \Lambda(\beta) \text{ where } \Lambda(\beta) := \ln \mathbb{E}[e^{\beta(X - \mathbb{E}[X])}].
$$

**Lemma 3** (Upper bound on conditional expectation). *Let $X : \Omega \to \mathbb{R}$ be a random variable on $(\Omega, \Sigma, \mathbb{P})$ such that the (restricted) cumulant generating function $\Lambda : \mathbb{R}^+ \to [0, \infty) \quad \beta \mapsto \ln \mathbb{E}[\exp(\beta(X - \mathbb{E}[X]))]$ exists. Assume further that $\mathbb{P}[\mathcal{B}] > 0$ such that $\mathbb{P}[\cdot|\mathcal{B}]$ is well-defined. Then with $\Lambda^*$ the Cramer rate function of $X$ (see Definition 1) it holds that*

$$
\mathbb{E}[X|\mathcal{B}] \leq \mathbb{E}[X] + (\Lambda^*)^{-1}(-\ln \mathbb{P}[\mathcal{B}]).
$$

*Proof.* The provided proof is an adaptation of Lemma 11 by Tarbouriech et al. (2024). We apply Lemma 1 with $p(\mathcal{E}) = \mathbb{P}[\mathcal{E}|\mathcal{B}]$ and $q(\mathcal{E}) = \mathbb{P}[\mathcal{E}]$, and restrict the supremum over just the random variables $\{\lambda(X - \mathbb{E}X)\}_{\lambda \in \mathbb{R}_+}$. This gives

$$
\begin{aligned}
D_{KL}[\mathbb{P}[\cdot|\mathcal{B}]||\mathbb{P}] &\geq \sup_{\lambda \in \mathbb{R}_+} \{\lambda\mathbb{E}[X - \mathbb{E}[X] \mid \mathcal{B}] - \ln \mathbb{E}[\exp(\lambda(X - \mathbb{E}X))]\} \\
&= \sup\{\lambda(\mathbb{E}[X|\mathcal{B}] - \mathbb{E}[X]) - \Lambda(\lambda) : \lambda \in \mathbb{R}_+\} \\
&= \Lambda^*(\mathbb{E}[X|\mathcal{B}] - \mathbb{E}[X])
\end{aligned}
$$

Furthermore, since $\lambda \in \mathbb{R}^+$ it follows that $\Lambda^*$ is strictly increasing and thus admits a strictly increasing inverse, i.e.,

$$
(\Lambda^*)^{-1}(D_{KL}[\mathbb{P}[\cdot|\mathcal{B}] \,||\, \mathbb{P}]) \geq \mathbb{E}[X|\mathcal{B}] - \mathbb{E}[X].
$$

As a final step, we use Lemma 2, which states that $D_{KL}[\mathbb{P}[\cdot|\mathcal{B}] \,||\, \mathbb{P}] = -\ln \mathbb{P}[\mathcal{B}]$. $\square$

**Corollary 1** (Upper bound on Gaussian conditional expectation). *Let $X \sim \mathcal{N}(\mu, \sigma^2)$ and $\mathbb{P}[\mathcal{B}] > 0$. Then*

$$
\mathbb{E}[X|\mathcal{B}] \leq \mu + \sigma\sqrt{-2\ln \mathbb{P}[\mathcal{B}]}.
$$

*Proof.* Since $X \sim \mathcal{N}(\mu, \sigma^2)$, it has a cumulant generating function $\Lambda(\beta) = \sigma^2\beta^2/2$ and a (Cramér) rate function $\Lambda^*(s) = s^2/(2\sigma^2)$ with inverse $(\Lambda^*)^{-1}(t) = \sigma\sqrt{2t}$. Plugging into Lemma 3 results in the desired expression. $\square$

**Corollary 2.** *Let $R \sim \mathcal{N}(\mu, K)$ with $\sigma_x := \sqrt{K_{x,x}}$. Define $R^* := \max_x R_x$ and $p^* := \mathbb{P}[R_x = R^*]$. Then*

$$
\mathbb{E}[R^*] \leq \mathcal{V}(p^*) \leq \max_{p \in \Delta^{|X|-1}} \mathcal{V}(p).
$$

*Proof.* The proof is a direct consequence of Corollary 1:

$$
\mathbb{E}[R^*] = \sum_x p_x^* \, \mathbb{E}[R_x|R_x = R^*] \leq \sum_x p_x^*(\mu_x + \sigma_x\sqrt{-2\ln p_x^*}).
$$

$\square$

**Lemma 4.** *Let $R : \Omega \to \mathbb{R}^{|X|}$ be a multivariate Gaussian with $\sigma_{R_x} \leq 1 \; \forall x \in X$ that is consecutively evaluated at locations $L^T := (x^t)_{t=1}^T$ with observations $Y_x = R_x + \epsilon_x$. The independent noise is distributed as $\epsilon_x \sim \mathcal{N}(0, \sigma_n^2(x))$ for $\sigma_n(x) \leq \sigma_n \; \forall x \in X$. Then the maximum information gain $\gamma^T$ upper bounds the aggregated predictive variances at the locations, i.e.,*

$$\gamma^T := \max_{L^T} I(Y_{L^T}; R) \geq 2 \underbrace{\frac{\ln(1+\sigma_n^{-2})}{4}}_{=:1/C_{\sigma_n}} \sum_{t=1}^T \sigma_{R_{x^t}^{t-1}}^2.$$

*Proof.* The proof generalizes that of Srinivas et al. (2010) from homoscedastic to heteroscedastic noise. First, note that the expression on the right hand side is well-defined, because $\sigma_{R_{x^t}^{t-1}}$ only depends on the observation locations $L^T$, but not on the observed value. Next, note that $Y_{L^T}|R$ is a multivariate normal with independent components of variance $\sigma_n^2(x^1), \ldots, \sigma_n^2(x^T)$, thus

$$I(Y_{L^T}; R) = H[Y_{L^T}] - H[Y_{L^T}|R]$$
$$= H[Y_{L^T}] - \tfrac{1}{2} \sum_{t=1}^T \ln(2\pi e \sigma_n^2(x^t)).$$

Furthermore, one may decompose

$$H[Y_{L^T}] = H[Y_{L^{T-1}}] + H[Y_{x^T}|Y_{L^{T-1}}]$$
$$= H[Y_{L^{T-1}}] + \tfrac{1}{2} \ln(2\pi e(\sigma_n^2(x^T) + \sigma_{R_{x^T}^{T-1}}^2)),$$

using that $Y_{x^T}|Y_{L^{T-1}}$ is Gaussian with variance $\sigma_n^2(x^T) + \sigma_{R_{x^T}^{T-1}}^2$. Recursively expanding then results in

$$I(Y_{L^T}; R) = \tfrac{1}{2} \sum_{t=1}^T \ln(1 + \sigma_n^{-2}(x^t) \sigma_{R_{x^t}^{t-1}}^2).$$

Finally, by assumption $\sigma_{R_{x^t}^{t-1}} \in [0, 1]$, allowing to lower bound each summand

$$\sigma_{R_{x^t}^{t-1}}^2 \leq \tfrac{1}{2} \ln(1 + \sigma_n^{-2}(x^t) \sigma_{R_{x^t}^{t-1}}^2) \tfrac{1}{2} \underbrace{\frac{4}{\ln(1+\sigma_n^{-2}(x^t))}}_{=:C_{\sigma_n(x^t)}}.$$

Indeed, the inequality is tight for $\sigma_{R_{x^t}^{t-1}} \in \{0, 1\}$ and holds in-between due to concavity of

$$\sigma_{R_{x^t}^{t-1}}^2 \mapsto \ln(1 + \sigma_n^{-2}(x^t) \sigma_{R_{x^t}^{t-1}}^2) - \sigma_{R_{x^t}^{t-1}}^2 \ln(1 + \sigma_n^{-2}(x^t)).$$

$\square$

**Proposition 3.** *Consider the 'Reinforce Leave One Out' (Kool et al., 2019) advantage function given by $\hat{r}_i - \frac{1}{B-1} \sum_{j \neq i} \hat{r}_j$ for i.i.d. samples $\hat{r}_j$ with $j \in \{1, \ldots, B\}$. Then the expected advantage is zero and an unbiased estimator of the variance is $\frac{1}{B} \sum_h (\hat{r}_h - \frac{1}{B-1} \sum_{l \neq h} \hat{r}_l)^2$. Moreover, standardizing RLOO results precisely in the advantage function employed by Shao et al. (2024) for 'Group Relative Policy Optimization' (GRPO):*

$$\underbrace{\frac{\hat{r}_i - \frac{1}{B-1} \sum_{j \neq i} \hat{r}_j}{\sqrt{\frac{1}{B} \sum_h (\hat{r}_h - \frac{1}{B-1} \sum_{l \neq h} \hat{r}_l)^2}}}_{\text{standardized RLOO}} = \underbrace{\frac{\hat{r}_i - \frac{1}{B} \sum_j \hat{r}_j}{\sqrt{\frac{1}{B} \sum_h (\hat{r}_h - \frac{1}{B} \sum_l \hat{r}_l)^2}}}_{\text{GRPO}}$$

*Proof of Proposition 3.* With linearity of the expectation and $\hat{r}_i \overset{i.i.d.}{\sim} p(\hat{r})$, the expected advantage is

$$\mathbb{E}[\hat{r}_i - \tfrac{1}{B-1} \sum_{j \neq i} \hat{r}_j] = \mathbb{E}[\hat{r}] - \tfrac{1}{B-1} \sum_{j \neq i} \mathbb{E}[\hat{r}] = 0 \quad \forall i.$$

From the same two properties, an unbiased estimator of the variance of the advantage also follows:

$$\mathbb{E}[\tfrac{1}{B} \sum_h (\hat{r}_h - \tfrac{1}{B-1} \sum_{l \neq h} \hat{r}_l)^2] = \mathbb{E}[(\hat{r}_h - \tfrac{1}{B-1} \sum_{l \neq h} \hat{r}_l)^2] - 0^2$$
$$= \text{Var}[\hat{r}_i - \tfrac{1}{B-1} \sum_{j \neq i} \hat{r}_j].$$

Finally, simple algebra results in equivalence between variance-regularized RLOO and GRPO:

$$\frac{\hat{r}_i - \frac{1}{B-1}\sum_{j\neq i}\hat{r}_j}{\sqrt{\frac{1}{B}\sum_h(\hat{r}_h - \frac{1}{B-1}\sum_{l\neq h}\hat{r}_l)^2}} = \frac{(B\hat{r}_i - \sum_j\hat{r}_j)/(B-1)}{\sqrt{\frac{1}{B}\sum_h((B\hat{r}_h - \sum_l\hat{r}_l)/(B-1))^2}}$$

$$= \frac{\frac{B}{B-1}(\hat{r}_i - \frac{1}{B}\sum_j\hat{r}_j)}{\sqrt{(\frac{B}{B-1})^2\frac{1}{B}\sum_h(\hat{r}_h - \frac{1}{B}\sum_l\hat{r}_l)^2}}$$

$$= \frac{\hat{r}_i - \frac{1}{B}\sum_j\hat{r}_j}{\sqrt{\frac{1}{B}\sum_h(\hat{r}_h - \frac{1}{B}\sum_l\hat{r}_l)^2}}.$$

$\square$

**Proposition 4.** *Let* $Y \sim \mathcal{N}(\nu, \lambda^2\Sigma)$ *and* $\mathbb{1} = (1, \ldots, 1)^T \in \mathbb{R}^s$. *Then for* $y \in \mathbb{R}^s$, *the (marginal) likelihood* $p(Y = y)$ *is maximized if*

$$\nu^{opt} = \frac{y^T\Sigma^{-1}\mathbb{1}}{\mathbb{1}^T\Sigma^{-1}\mathbb{1}} \text{ and } \lambda^{opt} = \sqrt{\frac{(y - \nu^{opt}\mathbb{1})^T\Sigma^{-1}(y - \nu^{opt}\mathbb{1})}{s}}.$$

*Proof of Proposition 4.* For $y = \alpha\mathbb{1}$, we get $\nu^{opt} = \alpha$ and $\lambda^{opt} = 0$, i.e., $Y = \alpha\mathbb{1}$ and $\mathbb{P}[Y = y] = 1$, which is clearly optimal. So, w.l.o.g. assume $y \not\propto \mathbb{1}$. Since $p(y)$ is a smooth function of $\nu \in \mathbb{R}, \lambda \in \mathbb{R}_+$, and because for any $y \not\propto \mathbb{1}$ it holds that $p(y) \to 0$ as $\nu \to \pm\infty$ or $\lambda \to \{0, \infty\}$, it suffices to consider the first-order conditions to find the maximizers. With Lemma 5 providing the gradients, the closed-form solutions follow swiftly. For $\nu$ we have

$$\frac{d}{d\nu} - 2\ln p(y) = -2(y - \nu\mathbb{1})^T\frac{1}{\lambda^2}\Sigma^{-1}\mathbb{1} = 0.$$

Likewise, for $\lambda$ we get

$$\frac{d}{d\lambda} - 2\ln p(y) = \frac{2}{\lambda}\text{Tr}((\frac{1}{\lambda^2}\Sigma^{-1} - \frac{1}{\lambda^2}\Sigma^{-1}(y - \nu\mathbb{1})(\frac{1}{\lambda^2}\Sigma^{-1}(y - \nu\mathbb{1}))^T)\lambda^2\Sigma)$$

$$= \frac{2}{\lambda}\text{Tr}(I - \frac{1}{\lambda^2}\Sigma^{-1}(y - \nu\mathbb{1})(y - \nu\mathbb{1})^T)$$

$$= \frac{2}{\lambda}(s - (y - \nu\mathbb{1})^T\Sigma^{-1}(y - \nu\mathbb{1})/\lambda^2) = 0.$$

$\square$

**Lemma 5.** *Let* $Y \sim \mathcal{N}(\mu_\phi, \Sigma_\psi)$ *with* $\phi = (\phi_1, \ldots, \phi_m)$ *and* $\psi = (\psi_1, \ldots, \psi_n)$. *Then*

$$-2\ln p(y) = \ln|\Sigma_\psi| + (y\text{-}\mu_\phi)^T\Sigma_\psi^{-1}(y\text{-}\mu_\phi) + const. \tag{7}$$

*Moreover, its gradients with respect to* $\phi_i$ *are*

$$-2(y - \mu_\phi)^T\Sigma_\psi^{-1}\frac{d}{d\phi_i}\mu_\phi \tag{8}$$

*and its gradients with respect to* $\psi_i$ *are*

$$\text{Tr}((\Sigma_\psi^{-1} - \Sigma_\psi^{-1}(y - \mu_\phi)(\Sigma_\psi^{-1}(y - \mu_\phi))^T)\frac{d}{d\psi_i}\Sigma_\psi). \tag{9}$$

*Proof of Lemma 5.* Equation (7) follows immediately by taking the logarithm of the multivariate normal probability density function, given by

$$p(y) = \frac{1}{\sqrt{(2\pi)^d|\Sigma_\psi|}}\exp(-(Y - \mu_\phi)^T\Sigma_\psi^{-1}(Y - \mu_\phi)/2).$$

Equation (8) follows from Equation (7) since

$$\frac{d}{d\phi_i} - 2\ln p(y) = \text{Tr}(\Sigma_\psi^{-1}\frac{d}{d\phi_i}(y - \mu_\phi)(y - \mu_\phi)^T)$$

$$= -2\text{Tr}(\Sigma_\psi^{-1}(y - \mu_\phi)\frac{d}{d\phi_i}\mu_\phi^T)$$

$$= -2(y - \mu_\phi)^T\Sigma_\psi^{-1}\frac{d}{d\phi_i}\mu_\phi,$$

where we have used symmetry of $\Sigma_\psi^{-1}$ in the last step. Likewise, Equation (9) results from Equation (7) through the following chain of equalities:

$$
\begin{aligned}
\frac{d}{d\psi_i} - 2\ln p(y) =& \mathrm{Tr}(\Sigma_\psi^{-1}\frac{d}{d\psi_i}\Sigma_\psi) + \mathrm{Tr}(\frac{d}{d\psi_i}\Sigma_\psi^{-1}(y-\mu_\phi)(y-\mu_\phi)^T) \\
=& \mathrm{Tr}(\Sigma_\psi^{-1}\frac{d}{d\psi_i}\Sigma_\psi - \Sigma_\psi^{-1}\frac{d\Sigma_\psi}{d\psi_i}\Sigma_\psi^{-1}(y-\mu_\phi)(y-\mu_\phi)^T) \\
=& \mathrm{Tr}(\Sigma_\psi^{-1}\frac{d}{d\psi_i}\Sigma_\psi - \Sigma_\psi^{-1}(y-\mu_\phi)(y-\mu_\phi)^T\Sigma_\psi^{-1}\frac{d\Sigma_\psi}{d\psi_i}) \\
=& \mathrm{Tr}((\Sigma_\psi^{-1} - \Sigma_\psi^{-1}(y-\mu_\phi)(\Sigma_\psi^{-1}(y-\mu_\phi))^T)\frac{d}{d\psi_i}\Sigma_\psi),
\end{aligned}
$$

where we have used that $\frac{d}{dx}|M| = |M| \cdot \mathrm{Tr}(M^{-1}\frac{d}{dx}M)$ and that $\frac{d}{dx}M^{-1} = -M^{-1}\frac{dM}{dx}M^{-1}$ for $M$ a matrix-valued function of $x$, as well as symmetry of $\Sigma_\psi^{-1}$. $\qquad\square$

## F   USE OF LARGE LANGUAGE MODELS

Large language models were used to aid and polish writing, in particular by giving feedback on the clarity of this paper, and to a minor extent for retrieval and discovery (i.e., finding related work). No large language model generated content was directly included without further adjustments.

