# OpenReview forum: "Thompson Sampling via Fine-Tuning of LLMs"
_ICLR.cc/2026/Conference — ICLR 2026 Poster_

### Official Review · Reviewer_9Xyb · 2025-10-19

**Soundness:** 3
**Presentation:** 3
**Contribution:** 3
**Rating:** 6
**Confidence:** 4

**Summary:**

This paper addresses Bayesian optimization in large unstructured discrete spaces by proposing ToSFiT (Thompson Sampling via Fine-Tuning), which parameterizes the probability of maximality (PoM) directly using large language models instead of performing expensive acquisition function maximization. The authors establish a novel regret bound for variational Thompson sampling that scales with the maximal information gain rather than the domain size, and demonstrate their approach on three diverse tasks: FAQ refinement, protein design, and quantum circuit optimization.

**Strengths:**

- Addresses a genuine problem: acquisition function maximization in large discrete spaces is prohibitively expensive
- The combination of pre-trained LLMs with principled Bayesian optimization is timely and well-motivated
- The improved regret bound is a solid theoretical contribution that advances understanding of VBOS
- Demonstrates the approach on three genuinely different domains (language, proteins, quantum circuits)

**Weaknesses:**

-  The performance of TOSFIT is fundamentally tied to the quality of the fixed feature map (i.e., the embeddings from a pre-trained model). If the pre-trained embeddings do not capture the features relevant to the reward function, the GP model may struggle, limiting the overall performance regardless of the policy optimization.
- Missing comparisons to other recent LLM-based optimization methods (e.g., FIBO is discussed but not compared experimentally), VAE-based methods, or even simpler approaches like evolutionary strategies
- Missing comparisons to other discrete optimization approaches (e.g., evolutionary methods, MCMC-based optimization)
- Limited discussion of when you would vs. wouldn't use this method. When is the 19% overhead worth it vs. just generating more candidates?

**Questions:**

- Why limit POST-GENERATION TS to 1000 candidates? What happens with 10k or 100k candidates with the same computational budget as ToSFiT?
- All experiments use small models (1.5-1.7B parameters). How does this scale to larger models?
- How does the method scale to longer sequence generation tasks?
- When does TOSFIT fail? When the LLM prior is misaligned, is it possible that TOSFIT will perform worse than random search?
- Have you considered using the LLM to propose multiple diverse candidates per forward pass (e.g., via beam search or sampling), which might improve the gradient estimator?
- Could you use the GP uncertainty to modulate the learning rate or number of fine-tuning steps adaptively?

---

> ### Author Response · Authors · 2025-11-20
> **Part 1**
>
> We are glad that the reviewer recognizes the importance of the problem we address and timeliness of introducing a principled method for LLM-driven Bayesian optimization. We are pleased that the improvements in the regret bound of VBOS are considered solid theoretical contributions. Finally, we are glad that the reviewer appreciates the diversity of the tasks experimentally considered. In the following, we address the questions and weaknesses raised by the reviewer.
>
> - *Fixed feature map*: We agree with the reviewer on the downsides of using fixed feature maps, see limitations in our original submission. We again want to emphasize that the recent work [1] has demonstrated how to integrate adaptive feature maps into deep Bayesian optimization loops (though they do not solve the intractability of acquisition function maximization). As such, these advancements are orthogonal to our contribution, i.e., they can be combined in future work. Moreover, there is ample precedent of applying Bayesian optimization algorithms designed for Gaussian rewards with known kernels to alternative reward models such as deep ensembles, e.g., [2,3]. To better illustrate the theoretical advancements of this work, we emphasized the controlled/simpler setting of a well-behaved Gaussian process reward model.
>
> - *Additional baselines*: We agree with the reviewer that a more extensive comparison with baselines was missing in the initial submission. We conducted extensive further experiments, and now compare against two RL baselines: actor critic and soft actor critic. As suggested, we also provide comparisons with fully in-context Bayesian optimization (FIBO), as well as LLM-based and LLM-free evolutionary search. Finally, as requested, we repeated the experiments with larger LLMs. We are excited to report that ToSFiT consistently outperforms all seven baselines. With these additions, we believe to have thoroughly covered the landscape of relevant discrete optimization methods, and strengthened our performance claims.
>
> - *Limited discussion on when to use method and 19\% overhead*: We thank the reviewer for their suggestion and have added a dedicated Table 1 to discuss the qualitative differences between ToSFiT and all baselines, guiding any practitioner. Moreover, we have demonstrated in Figure 7a that ToSFiT compensates for the compute overhead with much better sample efficiency, resulting in state-of-the-art computational efficiency. Note that these results in particular imply that at any compute budget, ToSFiT reaches higher rewards than POST-GENERATION TS.

---

> ### Author Response · Authors · 2025-11-20
> **Part 2**
>
> *Additional Questions*:
> - As discussed above and made more clear in the updated version of the paper, ToSFiT is more computationally efficient than POST-GENERATION TS. Figure 7a now directly compares the runtime of ToSFiT with that of UNGUIDED GENERATION, a theoretical upper bound on the computational efficiency of POST-GENERATION TS, for up to 10'000 steps. Note that the figure in particular demonstrates a widening performance gap between ToSFiT and UNGUIDED GENERATION/POST-GENERATION TS as the number of rounds is increased, due to the latter two wasting compute on a fixed model that does not take into account posterior information.
>
> - While the model size is small for LLM standards, it is in fact orders of magnitude larger than any previous demonstration of variational Bayesian optimistic sampling [4, 5]. In practice, running the experiments for 1000 steps to statistical significance (25 seeds) results in substantial compute cost. Nevertheless, we have added additional experimental results on $7B$ and $8B$ models to Figure 4. As reported by [6], scaling to larger models yields limited benefits on sample efficiency and incurs significantly more compute cost, a finding that our results confirm. As long as the model is able to roughly parameterize the manifold of valid solutions, optimization with principled exploration/exploitation tradeoff suffices to efficiently traverse the manifold.
>
> - The search space in longer generation tasks is naturally exponentially larger. Yet, in our structure-aware regret bound the dependency on the cardinality of the search space is at most logarithmic. Thus, we expect additional computational benefits compared to acquisition function maximization.
>
> - The primary failure point of ToSFiT occurs if the feature map (or ensemble etc) does not appropriately capture the structure of the reward, as was already mentioned by the reviewer. In contrast, misalignment of the LLM prior is not a statistical issue but simply requires further computational effort. This is reflected in the dependency of the regret bound on the policy prior only through the Bregman divergence between the policy and the VBOS maximizer.
>
> - Such variance reduction techniques in the style of importance sampling have come to mind, but were left to future work to simplify the first version of ToSFiT.
>
> - Ideally, the learning rate should be small initially while the signal from the reward model is still miscalibrated. The signal becomes more calibrated as the uncertainty on the reward model decreases, so this might be used to adaptively adjust the learning rate. We refer to future work for developing such heuristics, and if possible, provide theoretical justification.
>
> We thank the reviewer for their constructive feedback and hope that by addressing all of their concerns we have convinced them to recommend acceptance. We are open to discuss any remaining open points.
>
> [1] Ranković, Bojana, and Philippe Schwaller. “GOLLuM: Gaussian Process Optimized LLMs — Reframing LLM Finetuning through Bayesian Optimization.” arXiv, 2025, arxiv.org/abs/2504.06265.
>
> [2] Russo, Daniel, and Benjamin Van Roy. "Learning to optimize via posterior sampling." Mathematics of Operations Research 39.4 (2014): 1221-1243.
>
> [3] Lu, Xiuyuan, and Benjamin Van Roy. "Ensemble sampling." Advances in Neural Information Processing Systems 30 (2017).
>
> [4] O'Donoghue, Brendan, and Tor Lattimore. "Variational Bayesian optimistic sampling." Advances in Neural Information Processing Systems 34 (2021): 12507-12519.
>
> [5] Tarbouriech, Jean, Tor Lattimore, and Brendan O'Donoghue. "Probabilistic inference in reinforcement learning done right." Advances in Neural Information Processing Systems 36 (2023): 33687-33725.

---

> ### Comment · Reviewer_9Xyb · 2025-11-21
> **Reply to Authors' Rebuttal**
>
> I have read the rebuttal, and the authors have satisfactorily addressed my concerns. I appreciate their effort in clarifying parts of the paper and conducting additional experiments within the stipulated time. I am happy to increase my score.

---

> > ### Author Response · Authors · 2025-11-28
> >
> > We are glad to have satisfactorily addressed all of the reviewers concerns and sincerely thank them for increasing their score on the presentation as well as their overall score.

---

### Official Review · Reviewer_YiCY · 2025-10-27

**Soundness:** 3
**Presentation:** 2
**Contribution:** 3
**Rating:** 6
**Confidence:** 3

**Summary:**

This paper introduces TOSFIT, a scalable approach to Bayesian Optimization (BO) over large unstructured discrete domains. Instead of maximizing acquisition functions directly, the method parameterizes the probability of maximality (PoM) using a prompt-conditioned large language model (LLM). TOSFIT adapts the pre-trained policy toward the posterior PoM through online fine-tuning guided by a Variational Bayesian Optimistic Sampling (VBOS) objective. The paper derives new theoretical regret bounds that match those of standard Thompson Sampling (TS) and GP-UCB, extends the analysis to approximate VBOS, and demonstrates empirical gains in frequently-asked-questions (FAQ) response refinement, protein search, and quantum circuit design tasks.

**Strengths:**

The paper proposes a new bridge between BO and LLM, replacing acquisition maximization with posterior-aligned fine-tuning. The regret analysis is solid. In addition, TOSFIT consistently outperforms unguided LLM generation and static candidate-based TS.

**Weaknesses:**

1. While theoretically rigorous, some derivations (e.g., the gradient of VBOS and its concavity proof) are dense and may be inaccessible to a broader ICLR audience. A high-level intuition section would improve readability.
2. The comparison is limited to two relatively weak baselines (unguided LLM and post-generation TS). More direct comparisons to in-context optimization or gradient-based RL fine-tuning (e.g., RLHF or FIBO) would strengthen empirical claims.
3. It remains unclear whether performance improvements stem primarily from the fine-tuning process or from better use of GP-based uncertainty.
4. Although the paper claims negligible overhead, it would be helpful to report absolute runtime and fine-tuning cost relative to BO iterations.

**Questions:**

Please see Weaknesses.

---

> ### Author Response · Authors · 2025-11-20
>
> We are glad that the reviewer appreciates the bridge established between Bayesian optimization and LLM-centric inference. We are pleased that the reviewer values our improved regret analysis. In the following, we address the questions and weaknesses raised by the reviewer.
>
> - *Dense derivations*: We appreciate the reviewer's comments on clarity and presentation. We are aware that the derivations are dense, and have tried to address this by incorporating proof sketches at the beginning of Appendix E already in the initial submission. This might have been missed by the reviewer. We now clearly separate the proof sketches from the dense proofs with section headers, and are happy to incorporate any further suggestions to improve clarity.
>
> - *Limited comparison*: We agree with the reviewer that a more extensive comparison with baselines was missing in the initial submission. We conducted extensive further experiments, and now compare against two RL baselines: actor critic (GRPO) and soft actor critic (entropy regularized GRPO). As can be seen from the additional experiments in Figure 4, the optimism term of ToSFiT leads to more efficient exploration than both actor critic baselines despite extensive tuning of the entropy term coefficient for the baseline methods. This is because ToSFiT directs exploration to candidate solutions with high uncertainty/novelty, leveraging the principle of optimism in the face of uncertainty. As suggested, we also provide comparisons with fully in-context Bayesian optimization (FIBO), as well as LLM-based and LLM-free evolutionary search. Moreover, we repeated all experiments with larger LLMs. We are excited to report that ToSFiT consistently outperforms all seven baselines. With these additions, we believe to have thoroughly covered the landscape of relevant discrete optimization methods, and strengthened our performance claims on ToSFiT.
>
> - *Performance improvements*: Given the comparison against actor critic, which corresponds to ToSFiT except setting $\sigma_x = 0$, we can now safely say that ToSFiT substantially benefits from the use of GP-based uncertainty by directly leveraging optimism during fine-tuning.
>
> - *Negligible overhead*: We refer the reviewer to the measurements in Appendix A.1 (originally mentioned at the end of Section 5.2 of the main text), which in fact report a 19\% runtime overhead from the additional backward pass. The overhead is in practice lower than 100\% due to parallelizability of backpropagation across the sequence length, which does not hold for the autoregressive forward pass. To put more emphasis on the computational cost in the main text, we now also include the best-seen reward as a function of the algorithm's runtime on an NVIDIA A100. Despite computational overhead at each round, the sample efficiency improvements of ToSFIT suffice to also reach state-of-the-art computational efficiency.
>
> We thank the reviewer for their constructive feedback and hope that by addressing all of their concerns, and by significantly extending the experimental section, we have convinced them to recommend acceptance. We are open to discuss any remaining open points.

---

> ### Author Response · Authors · 2025-11-26
> **Reminder**
>
> Dear Reviewer,
>
> We hope this message finds you well. We have noticed that our detailed rebuttal, addressing each of the concerns raised in your review, has not yet received any feedback. We understand the demanding nature of the review process and appreciate the time and effort invested in evaluating our work.
>
> We kindly urge you to consider our responses to your questions, as we believe they adeptly address your concerns. With some days left, we hope we can still have a fruitful rebuttal period.
>
> Thank you, Authors

---

### Official Review · Reviewer_c3Vs · 2025-10-31

**Soundness:** 4
**Presentation:** 4
**Contribution:** 4
**Rating:** 8
**Confidence:** 4

**Summary:**

This paper tackles the prohibitive cost of Bayesian optimization (BO) in large, unstructured discrete spaces. The authors’ key move is to eliminate expensive acquisition-function maximization over huge domains by directly sampling the next candidate from a parameterized distribution that approximates the posterior probability of maximality (PoM). They instantiate Thompson Sampling as fine-tuning a large language model (tagged TOSFIT) using a prompt-conditioned, pre-trained LLM that is carefully adapted toward PoM. The work builds on Variational Bayesian Optimistic Sampling (VBOS; O’Donoghue & Lattimore, 2021), but particularly strengthens it; they reformulate the analysis to yield a structure-aware regret bound (in terms of information gain) and extend it to approximate VBOS, which directly motivates TOSFIT’s emphasis on pre-trained policy initialization and cautious, PoM-aligned adaptation.

**Strengths:**

1. The paper is well motivated by both the challenges of BO in large discrete settings and the limitations of existing approaches. The authors demonstrate a solid command of the literature.

2. The algorithm is well argued and clearly presented. Although the work builds on VBOS, the authors provide clear theoretical reformulations and insight driven analysis that justify the development of TOSFIT.

3. The experiments show clear performance gains over the chosen baselines.

4. The mathematical assumptions, propositions, and theorems are clearly stated and organized in the main paper and the supplementary material.

5. The scalability of the proposed methodology is well expounded, with clear guidelines for practice and future research directions.

**Weaknesses:**

1. The paper could give a clearer background on the core concept and regret formulations behind VBOS before introducing the intimate connection to UCB. While the adaptive UCB exploration bonus is central to the gradient formulation, the explicit bridge from VBOS to regret bounds could be surfaced earlier. VBOS exact regret bound via the optimistic set appears in Proposition 2, but new readers may struggle without an earlier signpost.

2. Fine tuning uses one VBOS gradient ascent step per round $c = 1$. This keeps cost low, but it raises natural questions about the compute versus quality trade off if more steps are used in complex settings.

Minor comment: The choice of a linear GP is well reasoned, but performance depends on the feature embedding. Using a fixed embedding in the experiments limits generalizability. The authors acknowledge this as future work and outline how to address it, which helps mitigate the concern.

**Questions:**

1. Regarding the choice of $c = 1$ for the fine tuning update, can the authors provide more detail on the compute versus performance trade off?
2. What guided the selection of baselines? Are there other established methods for combinatorial or discrete BO that were considered but not included? A brief rationale for the baseline set would be helpful.

---

> ### Author Response · Authors · 2025-11-20
>
> We are glad that the reviewer finds our paper well-motivated and that they agree it fills an important gap in the literature. We are pleased that the reviewer appreciates the clarity of our exposition and recognizes our strengthening of VBOS in terms of a structure-aware regret bound that also holds for approximate instantiations of VBOS, as well as the empirical performance gains achieved via ToSFiT. In the following, we address the questions and weaknesses raised by the reviewer.
>
> - *Exposition of VBOS*: We agree with the reviewer that VBOS warrants further exposition given its importance to the development of ToSFiT. Accordingly, we now extended its introduction, subject to page constraints. In particular, we highlight how VBOS leverages an upper bound on the expected maximum reward and how its (and our structure-aware) regret bounds hold for any policy in the optimistic set. Concretely, we extend the exposition of VBOS to include the following statement: "Moreover, $\max_{\pi} \mathcal V(\pi) \geq \mathcal V(\mathrm{PoM}) \geq \mathbb E[R^{\ast}]$ (see Corollary 2 in Appendix E), which is the key property that allows bounding the expected cumulative regret by $\tilde{\mathcal{O}}(\sqrt{T})$ for any optimistic policy $\pi$ with $\mathcal V(\pi) \geq \mathbb E[R^*]$". We are open for further discussion on this point if needed.
>
> - *Fine-tuning steps*: The reviewer correctly notes that in the initial submission we have emphasized the compute-efficient setup of only one step of fine-tuning per step of Bayesian optimization. Indeed, additional steps lead to a more sample-efficient optimization trajectory at a higher computational cost. To demonstrate this phenomenon, we now also include a plot in Figure 7b of Section 5.5 where the number of steps of gradient ascent per round c is swept across {2,4,8} for the protein optimization setup. Larger c result in better trajectories, though at a higher computational cost.
>
> Questions:
> - Q1) Addressed above.
> - Q2) The two initial baselines were selected according to their simplicity, generality, and scalability to large discrete settings. However, we acknowledge that a more exhaustive comparison against alternative methods is warranted to properly assess the capabilities of ToSFiT. To further strengthen our submission, we now also compare in Figure 4 against actor critic and soft actor critic. The optimism of ToSFiT leads to better optimization trajectories than either, despite extensive tuning of the entropy term coefficient in the baselines. This is because ToSFiT directs exploration to candidate solutions with high uncertainty/novelty, leveraging the principle of optimism in the face of uncertainty. Furthermore, we have extended Figure 4 with fully in-context Bayesian optimization (FIBO) as well as LLM-based and LLM-free evolutionary search. ToSFiT consistently outperforms all seven baselines, both when applied to the LLMs of the original submission, as well as when applied to larger LLMs included in the updated submission. With these additions, we believe to have thoroughly covered the landscape of relevant discrete optimization methods.
>
> We thank the reviewer for their constructive feedback and hope that by addressing all of their questions and concerns, as well as through the substantial extension of the empirical section establishing state-of-the-art performance for ToSFiT, they agree that our submission has been further improved. We are happy to discuss any open points.

---

> ### Author Response · Authors · 2025-11-26
> **Reminder**
>
> Dear Reviewer,
>
> We hope this message finds you well. We have noticed that our detailed rebuttal, addressing each of the concerns raised in your review, has not yet received any feedback. We understand the demanding nature of the review process and appreciate the time and effort invested in evaluating our work.
>
> We kindly urge you to consider our responses to your questions, as we believe they adeptly address your concerns. With some days left, we hope we can still have a fruitful rebuttal period.
>
> Thank you, Authors

---

### Official Review · Reviewer_UTrs · 2025-11-01

**Soundness:** 3
**Presentation:** 2
**Contribution:** 3
**Rating:** 4
**Confidence:** 3

**Summary:**

This paper proposes a variational Bayesian optimization surrogate (VBOS) objective that provides a tighter bound for Thompson sampling–based optimization. The method enables gradient-based fine-tuning of generative models under non-differentiable and expensive reward functions.

**Strengths:**

The paper presents an interesting and theoretically grounded idea. The VBOS formulation provides a principled way to incorporate uncertainty into generative model fine-tuning for non-differentiable rewards. The theoretical analysis is sound and detailed.

**Weaknesses:**

- GP: The paper relies on the assumption that the underlying reward follows a Gaussian process prior. Will this be held for discrete or highly non-smooth reward functions?
- Baselines: while the paper focuses on Bayesian optimization–based baselines, it omits comparisons with standard reinforcement learning (RL) methods such as PPO/GRPO, which can also handle non-differentiable rewards to fine-tune LLMs. (Are there any reasons that PPO/GRPO is not able to be used here?) Empirically, I can see VBOS adds an uncertainty term $\sigma_x$; however, such comparisons would be crucial to demonstrate that the exploration term can contribute beyond simple reward-based fine-tuning.
- Diversity measure: Reward-based fine-tuning sometimes leads to diversity collapse to a narrow high-reward set. Since the proposed VBOS objective explicitly includes an uncertainty term that should encourage exploration, it would be valuable to report how diversity changes during optimization.
- Performance convergence: In Figure 4, ToSEiF appears to achieve higher rewards given the same computational cost. However, it is unclear when the fine-tuning process converges. If I have enough resources, when to stop the fine-tuning? Could the authors provide more complete curves, extending beyond the current range, potentially into regions where ToSEiF’s performance even declines (perhaps in the appendix)? This would help readers better understand the convergence behavior and overall stability of the optimization process.

**Questions:**

Following the weakness
- For the given tasks, if apply RL methods such as PPO/GRPO, what would be the results? How your methods be better than RL? (or how your methods are correlated with RL?)
- What is the diversity measure during the fine-tuning? Do you have mode-collapse problem?
- How to select the convergence point, if I have enough computation resources?

---

> ### Author Response · Authors · 2025-11-20
>
> We are glad that the reviewer finds our algorithm interesting and appreciates its theoretical motivation, in particular with respect to uncertainty-aware exploration, as well as its detailed analysis. In the following, we address and clarify all questions and weaknesses raised.
>
> - *GP assumption*: While it is true that the regret bound presented assumes the standard setting of Gaussian rewards, we point out that (1) none of the reward functions in the experiments are Gaussian, (2) Bayesian optimization algorithms are usually first analyzed in the Bayesian setting [1] before extending the analysis to the frequentist setting [2]. In the frequentist setting, BO algorithms still assume Gaussianity algorithmically, but the regret bounds hold for any fixed reward function in some reproducing kernel Hilbert space. Such an extension should also be possible in our case, though as in [2], we expect the bound to be slightly looser. To simplify the exposition in an already theoretically dense paper, we restricted our analysis to the Gaussian case.
>
> - *Baselines*: We agree with the reviewer that comparison with standard RL techniques was missing. In the updated version, we also compare against actor critic (GRPO) and soft actor critic (entropy regularized GRPO). As can be seen from the additional experiments in Figure 4, the optimism term of ToSFiT leads to more efficient exploration than both actor critic baselines despite extensive tuning of the entropy term coefficient for the baseline methods. This is because ToSFiT directs exploration to candidate solutions with high uncertainty/novelty, leveraging the principle of optimism in the face of uncertainty. Moreover, we provide additional comparisons with fully in-context Bayesian optimization (FIBO) as well as LLM-based and LLM-free evolutionary search. Finally, we repeated all experiments with larger LLMs. We are excited to report that ToSFiT consistently outperforms all seven baselines. With these additions, we believe to have thoroughly covered the landscape of relevant discrete optimization methods.
>
> - *Diversity measure*: We thank the reviewer for their suggestion. We have now added an additional analysis on the diversity of candidate generation during Bayesian optimization in Section 5.3 of the main text (Figure 5) as well as a more exhaustive analysis in Section A.2 of the Appendix. As recognized by the reviewer, the uncertainty-directed entropy regularization of ToSFiT induce diversity, avoiding the diversity collapse that plagues standard reward-based fine-tuning. Moreover, compared to soft actor critic, diversity is adjusted in a more targeted and stable manner. Finally, although never observed for ToSFiT, in contrast to RL, mode collapse after sufficient exploration is not necessarily undesirable: indeed, in BO the model is discarded after optimization.
>
> - *Performance convergence*: We believe there might be a misunderstanding. Figure 4 reports on the sample efficiency, i.e., on the best-seen reward as a function of the number of interactions with the real-world environment. The best-seen reward can, by definition, never drop as more candidates are evaluated. Fine-tuning occurs only to minimize the Bregman divergence term in the regret bound of Theorem 1, in fact, ToSFiT obtains the strongest regret bounds by adopting an infinite number of fine-tuning steps per round of Bayesian optimization. This works because we fine-tune toward a probabilistic model of the reward rather than actual reward realizations, so no overfitting occurs. Empirically, we now confirm the benefits of additional fine-tuning in Figure 7b. Moreover, we now include a discussion on the (non-)risk of overfitting neural policies during Bayesian optimization in Section A.3 of the Appendix. What can be observed in Figure 4, is stagnation of the per-round improvements on the best-seen reward, either through a miscalibrated probabilistic reward model, or due to discovering a near-optimal candidate solution. As per the reviewer's suggestion, we now additionally report on 10x more rounds of Bayesian optimization with Figure 7a.
>
> *Questions*: We hope that all of the reviewer's questions were sufficiently addressed in the exposition above and the changes made to the paper.
>
> We thank the reviewer for their constructive feedback and hope that by addressing all of their concerns and substantially extending the empirical validation of ToSFiT we have convinced them to recommend acceptance. We are happy to further discuss any open points.
>
> [1] Russo, Daniel, and Benjamin Van Roy. "Learning to optimize via posterior sampling." Mathematics of Operations Research 39.4 (2014): 1221-1243.
>
> [2] Chowdhury, Sayak Ray, and Aditya Gopalan. "On kernelized multi-armed bandits." International Conference on Machine Learning. PMLR, 2017.

---

> ### Author Response · Authors · 2025-11-26
> **Reminder**
>
> Dear Reviewer,
>
> We hope this message finds you well. We have noticed that our detailed rebuttal, addressing each of the concerns raised in your review, has not yet received any feedback. We understand the demanding nature of the review process and appreciate the time and effort invested in evaluating our work.
>
> We kindly urge you to consider our responses to your questions, as we believe they adeptly address your concerns. With some days left, we hope we can still have a fruitful rebuttal period.
>
> Thank you, Authors

---

### Author Response · Authors · 2025-11-20
**Rebuttal Global Response**

We are glad that the reviewers appreciate the timely bridge established between Bayesian optimization and LLM-centric inference [YiCY, 9Xyb], filling an important yet unaddressed gap in the literature [c3Vs, 9Xyb]. We are pleased that they appreciate the theoretical motivation of ToSFiT [UTrs, c3Vs] and value our theoretical contributions [YiCY, UTrs, c3Vs, 9XyB].

We are glad that reviewer 9Xyb recognizes the diversity of the problems considered for experimental evaluation. Nevertheless, we agree with reviewers [UTrs, YiCY, and 9Xyb] that our limited comparison against only two baselines made it difficult to assess the practical benefit of ToSFiT. We are excited to report that we have extended the empirical analysis to 7 baselines and 2 model sizes per experimental setting. Across all experiments, ToSFiT exhibits state-of-the-art sample efficiency as well as computational efficiency, sometimes by a large margin. The baselines against which we now compare are as follows:

1) Unguided Generation ([1], as before).
2) Post-Generation TS ([2,3], as before).
3) Actor Critic ([4], requested by reviewers UTrs and YiCY).
4) Soft Actor Critic ([5], requested by reviewers UTrs and YiCY).
5) Fully In-context Bayesian Optimization ([6], suggested by reviewers YiCY and 9Xyb).
6) LLM-based Evolutionary Search ([7], suggested by reviewer 9Xyb).
7) Classical Discrete Evolutionary Search ([8], suggested by reviewer 9Xyb).

Furthermore, we now report on the policy diversity during Bayesian optimization (suggested by reviewer UTrs), computational efficiency (suggested by reviewer 9Xyb, YiCY, and c3Vs), and scalability to larger models and compute budgets (suggested by reviewer 9Xyb and c3Vs).

To assist the reviewers with the updates made to the paper, we have marked all changes with blue font. We will of course revert back to black font upon acceptance.

We sincerely thank the reviewers for their constructive feedback, which greatly helped improve the paper. We hope that by addressing all of their individual concerns we have convinced them to raise their scores. We are open to discuss any remaining points.

[1] Li, Yujia, et al. "Competition-level code generation with alphacode." Science 378.6624 (2022): 1092-1097.

[2] Kristiadi, Agustinus, et al. "A sober look at LLMs for material discovery: Are they actually good for Bayesian optimization over molecules?." arXiv preprint arXiv:2402.05015 (2024).

[3] Ranković, Bojana, and Philippe Schwaller. "GOLLuM: Gaussian Process Optimized LLMs--Reframing LLM Finetuning through Bayesian Optimization." arXiv preprint arXiv:2504.06265 (2025).

[4] Barto, Andrew G., Richard S. Sutton, and Charles W. Anderson. "Neuronlike adaptive elements that can solve difficult learning control problems." IEEE transactions on systems, man, and cybernetics 5 (2012): 834-846.

[5] Haarnoja, Tuomas, et al. "Soft actor-critic: Off-policy maximum entropy deep reinforcement learning with a stochastic actor." International conference on machine learning. Pmlr, 2018.

[6] De Carvalho, Gustavo Sutter Pessurno, et al. "Simplifying Bayesian Optimization Via In-Context Direct Optimum Sampling." arXiv preprint arXiv:2505.23913 (2025).

[7] Romera-Paredes, Bernardino, et al. "Mathematical discoveries from program search with large language models." Nature 625.7995 (2024): 468-475.

[8] Holland, John H. "Genetic algorithms and adaptation." Adaptive control of ill-defined systems. Boston, MA: Springer US, 1984. 317-333.

---

### Author Response · Authors · 2025-12-01
**Summary of the Discussion Phase**

We thank the reviewers and ACs for their time and valuable feedback. The paper was received very positively by reviewer c3Vs (score 8), who particularly praised the soundness, presentation, and contribution. Reviewer 9Xyb initially requested additional baselines but, upon reviewing our new experiments, raised their score from 6 to 8. These two reviewers were both confident in their assessment (confidence 4).

Reviewers UTrs and YiCY (confidence 3) appreciated the theoretical contributions but requested broader empirical comparisons. We fully addressed these concerns by extending the baselines from 2 to 7 (covering classical BO, in-context BO, evolutionary search, and RL) and doubling the model size configurations. These results demonstrate that our method achieves state-of-the-art sample and computational efficiency while avoiding diversity collapse.

Regarding the discussion timeline: We posted our full rebuttal on November 20th. Reviewer 9Xyb responded immediately, raising their score. However, due to the unexpected OpenReview leak, the discussion period was curtailed, preventing the remaining reviewers from officially updating their scores or acknowledging the new results. We are confident that the extensive new experiments fully resolve the concerns raised by UTrs and YiCY.

---

### Meta-Review · Area_Chair_v2pv · 2026-01-05

**Summary:**

This paper proposes fine-tuning an LLM to approximate the probability of maximization (PoM) distribution of a Gaussian process for Bayesian optimization over discrete spaces. The authors ground their method in existing theory on variational Bayesian optimistic sampling, provide theoretical guarantees, and evaluate on many discrete benchmarks with comparisons against competitive methods. While there are some outstanding issues related to the presentation/accessibility of the methods section, and while there are noted limitations of the method (e.g., fixed feature maps, limited fine-tuning steps), overall, the strengths of this paper outweigh its weaknesses. The approach of variationally approximating the PoM distribution is novel, the use of LLMs to do so is timely, and the application of discrete-space BO problems is relevant. It is a high-quality, well-grounded, and rigorous method, and it will be a useful contribution to the ICLR conference and the BO community at large.

**Reviewer Concerns:**

There were two large themes of issues brought up during the reviewing process: (1) presentation/clarity and (2) baseline methods. The former was addressed by the author's comments, though many reviewers did not respond to whether specific concerns were resolved. The latter was addressed by the authors, including many more relevant baselines, including methods from the RL literature. Again, many reviewers did not indicate whether these changes were satisfactory; however, I deemed them sufficient to demonstrate the merit of this paper’s contributions.

**Reviewer Scores:**

More engagement from the reviewers could have helped or hurt this paper, depending on whether they found the additional clarity/benchmarks to improve or detract from it. Nevertheless, given the reviewers' initial enthusiasm and my own reading of the paper, I feel confident in recommending it for acceptance.

- Reviewers UTrs, YiCY, and c3Vs did not engage at all during the discussion, and so I’d suspect they wouldn’t have changed their scores with more discussion.
- Reviewer 9Xyb acknowledge that their concerns were addressed and increased their score. Therefore, more discussion would likely not have further changed their mind.

---

### Decision · Program_Chairs · 2026-01-26

Accept (Poster)